# Global transcription factors analyses reveal hierarchy and synergism of regulatory networks and master virulence regulators in *Pseudomonas aeruginosa*

Jiadai Huang[1†], Yue Sun[1†], Fang Chen[1†], Shumin Li[2], Xiangkai You[1], Liangliang Han[1], Jingwei Li[1], Zhe He[1], Canfeng Hua[1], Chunyan Yao[1], Tianmin Li[1], Beifang Lu[1], Yung-Fu Chang[3], Xin Deng[1,4]*

[1]Department of Biomedical Sciences, City University of Hong Kong, Hong Kong, China; [2]Department of Computer Science, The University of Hong Kong, Hong Kong, China; [3]Department of Population Medicine and Diagnostic Sciences, College of Veterinary Medicine, Cornell University, Ithaca, United States; [4]Shenzhen Research Institute, City University of Hong Kong, Shenzhen, China

*For correspondence:
xindeng@cityu.edu.hk

[†]These authors contributed equally to this work

Competing interest: The authors declare that no competing interests exist.

## eLife Assessment

This study provides an **important**, comprehensive, large-scale dataset on transcription factor binding in *Pseudomonas aeruginosa*, along with analyses of its regulatory network, key virulence and metabolic regulators, and a pangenomic examination of transcription factors. Utilizing large-scale ChIP-seq and multi-omics integration, the research **convincingly** supports the hierarchical regulatory structures and offers insights into virulence mechanisms. This dataset, made available through an online database, should be an invaluable resource to the research community studying *P. aeruginosa*, a key pathogen at risk for hospital infections and development of antibiotic resistance.

**Abstract** The transcription factor (TF) regulatory network in *Pseudomonas aeruginosa* is complex and involves multiple regulators that respond to various environmental signals and physiological cues by regulating gene expression. However, the biological functions of at least half of its 373 putative TFs remain uncharacterised. Herein, chromatin immunoprecipitation sequencing (ChIP-seq) was used to investigate the binding sites of 172 TFs in the *P. aeruginosa* PAO1 strain. The results revealed 81,009 significant binding peaks in the genome, more than half of which were located in the promoter regions. To further decode the diverse regulatory relationships among TFs, a hierarchical network was assembled into three levels: top, middle, and bottom. Thirteen ternary regulatory motifs revealed flexible relationships among TFs in small hubs, and a comprehensive co-association atlas was established, showing the enrichment of seven core associated clusters. Twenty-four TFs were identified as the master regulators of virulence-related pathways. The pan-genome analysis revealed the conservation and evolution of TFs in *P. aeruginosa* complex and other species. A web-based database combining existing and new data from ChIP-seq and the high-throughput systematic evolution of ligands by exponential enrichment was established for searching TF-binding sites. This study provides important insights into the pathogenic mechanisms of *P. aeruginosa* and related bacteria and is expected to contribute to the development of effective therapies for infectious diseases caused by this pathogen.

## Introduction

Transcription factors (TFs) are key regulators of transcription and thus play crucial roles in mediating multiple biological pathways and events in eukaryotes and prokaryotes. Based on their DNA-binding domains (DBDs), TFs can be classified into different families, such as the LysR, AraC, and LuxR families (*El-Gebali et al., 2019*). TFs can activate and repress the expression of a set of genes in response to various environmental and/or specific signal triggers (*Salgado et al., 2007*; *Seshasayee et al., 2011*). Several approaches can be used to study TFs, including chromatin immunoprecipitation sequencing (ChIP-seq) (*Johnson et al., 2007*) and ChIP-exo (*Rhee and Pugh, 2011*) in *in vivo* research and DNA affinity purification sequencing (DAP-seq) (*Bartlett et al., 2017*) and high-throughput systematic evolution of ligands by exponential enrichment (HT-SELEX) (*Jolma et al., 2013*) in *in vitro* research. ChIP-seq is a powerful *in vivo* technique that enables more accurate genome-wide mapping of TF-DNA interactions than *in vitro* methods because TFs may interact with other co-regulators in an environment-specific pattern, thereby altering binding preferences *in vivo* (*Furey, 2012*; *Fu et al., 2022*).

These sequencing approaches have been used to illustrate the biological functions of TFs in several bacterial species, including *Mycobacterium tuberculosis*, *Vibrio cholerae*, *Salmonella enterica*, *Escherichia coli*, and *Clostridium thermocellum* (*Minch et al., 2015*; *Ayala et al., 2017*; *García-Pastor et al., 2019*; *Seo et al., 2014*; *Ishihama et al., 2016*; *Hebdon et al., 2021*). In addition, several databases and models have been constructed based on experimental validation or machine learning for studying TFs in bacteria, such as CollecTF, PredicTF, PRODORIC, DBTBS, and RegulomePA (*Kiliç et al., 2014*; *Oliveira Monteiro et al., 2022*; *Dudek and Jahn, 2022*; *Makita et al., 2004*; *Galán-Vásquez et al., 2020*). However, the functions and spatiotemporal interactions of most bacterial TFs remain to be clarified.

The gram-negative bacterium *Pseudomonas aeruginosa* is a major human opportunistic pathogen that grows ubiquitously. *P. aeruginosa* tends to cause infection in burn victims, patients with cystic fibrosis (CF), and individuals hospitalised for long periods (*Bodey et al., 1983*; *Stover et al., 2000*). It utilises versatile pathways to exert its virulence, such as biofilm formation, quorum sensing (QS), type III and type VI secretion systems (T3SS and T6SS, respectively), motility, siderophore production, oxidative stress resistance, and antibiotic resistance, all of which are under the control of a complicated TF regulatory network (TRN). The metabolic changes regulated by the TRN enhance the pathogen's adaptability under different infection and survival conditions.

*P. aeruginosa* employs diverse virulence pathways to establish successful infection, with QS being one of the major mechanisms involving the expression of many virulence genes. Its regulation occurs hierarchically, comprising the interconnected *las*, *rhl*, and *pqs* systems (*Lee and Zhang, 2015*; *Girard and Bloemberg, 2008*). Through its transcriptional regulator, LasR, and in collaboration with its corresponding autoinducer, 3-oxo-$C_{12}$-HSL (encoded by *lasI*), the *las* system triggers the activation of the *rhl* and *pqs* systems (*Lee and Zhang, 2015*; *Schuster and Greenberg, 2007*). Additionally, *P. aeruginosa* utilises motility systems for surface attachment and colonisation, including (1) swarming and swimming powered by flagella, which are multicellular and individual cell movements, respectively, and (2) twitching powered by type 4 pili (T4P) (*Kazmierczak et al., 2015*), wherein T4P promotes surface attachment by twitching to pull the cell closer to the attachment sites (*Kearns, 2010*). The motility of *P. aeruginosa* is thus the initial step in its pathogenic process and is associated with the upregulation of virulence and the induction of host defence. Both QS and motility pathways are precisely controlled by the sophisticated TRN in *P. aeruginosa*. Understanding how these virulence pathways are integrated and regulated is crucial for developing novel therapeutic strategies against *P. aeruginosa* infections.

Despite the TRN being essential for understanding how bacterial genes are regulated synergistically, only a limited number of binding targets of TFs have been identified in *P. aeruginosa* over the past decades. For example, ChIP-seq and RNA sequencing (RNA-seq) co-analysis has been used to map a *P. aeruginosa* genomic regulatory network (PAGnet) of 20 key virulence-related TFs, while HT-SELEX has been used to characterise the DNA-binding specificities of 182 TFs (*Wang et al., 2021*). DAP-seq has been used to reveal a regulatory network of 55 response regulators (RRs) of two-component systems in *P. aeruginosa*, 51 of which are from the PAO1 strain (*Trouillon et al., 2021*). However, almost half of the TFs remain functionally uncharacterised, and their downstream targets and upstream regulators are unknown. To fill this knowledge gap, we performed ChIP-seq of 172 TFs, representing the majority of TFs not covered by HT-SELEX, to map their genomic binding sites *in vivo*

(*Supplementary file 1*). This comprehensive dataset revealed a hierarchical and co-association regulatory network among TFs and newly identified virulence-related regulators. We integrated both the ChIP-seq and HT-SELEX data into a web-based database for querying TF-binding patterns, which is publicly available. Our findings and tools will significantly contribute to future mechanistic studies on and a comprehensive understanding of the TRN in *P. aeruginosa* and other bacterial species.

## Results

### ChIP-seq of 172 uncharacterised TFs reveals a global transcriptional atlas of *P. aeruginosa*

In this study, 373 proteins in *P. aeruginosa* were annotated as TFs with DNA-binding activity based on the existing annotations in the *Pseudomonas* Genome Database (*Winsor et al., 2016*; *Figure 1A*). To profile the whole TF-DNA-binding landscape and construct a comprehensive regulatory network of *P. aeruginosa*, ChIP-seq experiments were performed on 172 TFs to determine the TF-binding sites (TFBSs) (*Figure 1A*). Libraries were constructed using vesicular stomatitis virus glycoprotein tagging, and raw sequence reads were mapped to the PAO1 genome using bowtie2 (Version 2.3.4.1) (*Langmead et al., 2009*). Using MACS2 (*Zhang et al., 2008*), 81,009 significant binding peaks with a cut-off *P*-value of 0.001 were identified. Subsequently, the R package ChIPpeakAnno (*Zhu et al., 2010*) was used to define the peak locations and find the nearest genes. The peak location can be divided into six features: upstream, overlapStart, inside, overlapEnd, downstream, and includeFeature. The percentages of binding peak locations for each TF indicated that more than half of the TFs preferentially bind to the upstream (upstream intergenic regions) and overlapStart (overlapped with the transcription start sites [TSSs] of genes) regions (*Figure 1B*). Apart from these regions, some TFs bind to the inside regions (coding regions of genes). Few TFs were found to preferentially bind to the rest of the peak location features.

To further provide insights into the functional roles of different TFs in gene regulation, we analysed the peak distance to the TSS, which revealed several patterns (*Figure 1C*). First, most of the TFs exhibited peak distributions that were concentrated in the proximal region around the TSS. Some TFs, such as Cor_CI, showed a broader distribution of peak distance, indicating that these TFs might also be active at positions farther from the TSS. Furthermore, certain TFs, including RpiR and DeoR, displayed distinct bimodal or multimodal distributions, while other TFs, such as LysR and AraC, showed smoother and more narrowly focused distributions, suggesting their different binding preferences. The number of peaks of each TF and its DBD types is summarised in a treemap in *Figure 1D*. TFs in the LysR and AraC families accounted for almost half of all of the peaks. PA2718, classified in the MerR family, yielded the highest number (2480) of binding peaks, followed by the TF PA0756 (2080 binding peaks) from the OmpR family. Furthermore, eight TFs showed no more than 10 peaks, and the TFs PA4074 and PA4806 had only two peaks each. To preliminarily identify the different regulatory functions of TFs via their different binding preferences, we performed Gene Ontology (GO) functional enrichment analysis for each TF and presented the top 10 GO terms (*Figure 1E*). The significantly enriched GO terms (BH-adjusted $P < 0.05$) for all of the 172 TFs analysed by ChIP-seq are listed in *Supplementary file 2*, and these terms suggest divergent functions of the TFs. Of these functional categories, most were related to metabolism, such as transcription, translation, ribosome structure, and GTPase activity, while some of the functional categories were associated with virulence, including T4P, O antigen biosynthesis, and biofilm formation (*Figure 1E*).

### Hierarchical networks of TFs based on pairwise interactions

TF-coding genes can also serve as targets for other TFs, leading to interactions between the DNA-binding profiles of TFs. To clarify these relationships, lists of binding targets of the TFs obtained using ChIP-seq and HT-SELEX were used to assemble a system-level hierarchical regulatory network. After filtering by TFs that bind to promoter regions, 947 unique pairwise patterns between TFs were identified from a total of 13,375 promoter interactions.

We next defined a statistic index, $h$, based on the out-degree ($O$) and in-degree ($I$) of interactions to measure the direction and hierarchical level of TFs [$h = (O − I)/(O + I)$] in our established hierarchical regulatory network and thus identify hubs and information-flow bottlenecks (*Gerstein et al., 2012*). The $h$ value of each TF can reflect the direction and extent of information flow. A positive $h$ indicates

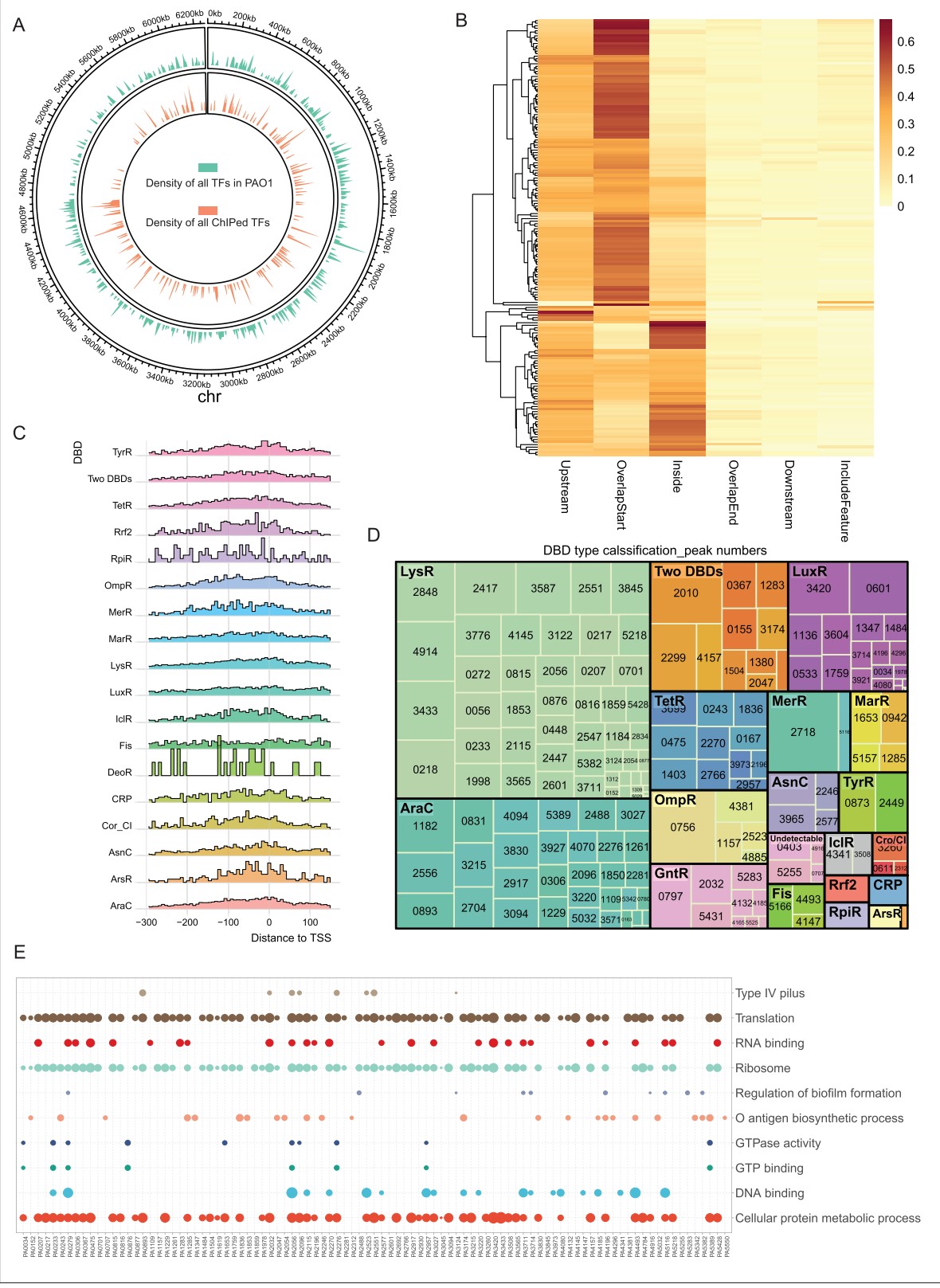

**Figure 1.** Overview of ChIP-seq results. (**A**) Density of all transcription factors (TFs, green) and ChIPed TFs (orange) in this study throughout the *P. aeruginosa* genome. (**B**) Annotation heatmap of all peak distribution with six locations: Upstream, where the peak is located entirely upstream of the gene; Downstream, where the peak is positioned completely downstream of the gene; Inside, where the peak is entirely contained within the gene body; OverlapStart, where the peak overlaps with the 5' end of the gene; OverlapEnd, where the peak overlaps with the 3' end of the gene;

*Figure 1 continued on next page*

*Figure 1 continued*

and IncludeFeature, where the peak completely encompasses the gene. (**C**) Peak distance to the translational start site (TSS) of each DBD family. (**D**) Treemap of the 172 TFs peak numbers based on DBD family. Each box's size represents the family's size (number of peaks), and the explained variance of each DBD type means the colour shades of each box. DBD families of ChIPed TFs are classified into 20 different categories: LuxR, LysR, Two DBDs (two DNA-binding domains), AraC, TetR, ArsR, CRP, OmpR, GntR, MarR, AsnC, Cro/CI, TyrR, Rrf2, MerR, IclR, Fis, RpiR, DeoR, and undetectable. (**E**) The dot plot shows the top 10 Gene Ontology (GO) terms from the PseudoCAP annotation of ChIPed TFs. The size of the dots indicates the significance of each functional category, quantified by $-\log_{10}$ (p. adjust).

that the TF acts 'upstream', whereas a negative value suggests that it acts 'downstream'. The higher the absolute $h$ value, the stronger the implication of the direction. A density plot of all of the TFs with their $h$ values revealed that they can be approximately divided into three groups: top ($h \in [0.75, 1]$), middle ($h \in [-0.75, 0.75]$), and bottom ($h \in [-1, 0.75]$) levels (*Figure 2—figure supplement 1A*). The top-level TFs are those that tend to control many other TFs, while the bottom-level TFs are those that are more regulated by other TFs than acting as regulators themselves. The middle-level TFs are those that connect the top- and bottom-level TFs. Of the 231 TFs in this hierarchy, 100, 46, and 85 TFs were defined as top-, middle-, and bottom-level TFs, respectively. The complete hierarchical information of all of the TFs is provided in *Supplementary file 3*. To avoid extreme cases and aid visualisation, we summarised the hierarchical regulatory network by removing TFs for which ($O + I$) was more than 10 (*Figure 2A* and *Figure 2—figure supplement 1B*). The network was visualised using Cytoscape software (*Shannon et al., 2003*).

Additionally, we investigated the factors influencing the hierarchical level of TFs, such as DBD types (e.g., DBD families containing more TFs). TFs from the AraC, GntR, and TetR families tended to cluster at the top level. TFs from the MarR family and two-DBD-type TFs (i.e., TFs with two DBDs) were mostly found in the bottom level. Additionally, among the characterised TFs, 58% were clustered at the bottom level, probably because these TFs directly regulated the downstream genes involved in important phenotypes, such as AlgR (*Lizewski et al., 2002*), AmrZ (*Jones et al., 2013*), MvfR (*Déziel et al., 2005*), FleQ (*Hickman and Harwood, 2008*), and VqsM (*Liang et al., 2014*). Overall, the organised hierarchical regulatory network profiling at the three levels showed complex pairwise interactions among TFs in *P. aeruginosa*.

## Ternary regulatory motifs show flexible relationships among TFs in *P. aeruginosa*

Apart from their global hierarchical regulatory structure, we investigated the constituent regulatory network motifs of TFs, which revealed small connectivity patterns associated with canonical functions (*Cheng et al., 2011*). Positive and negative feedback loops are prevalent in bacterial regulatory networks, which generally involve combinations of several genes to appropriately respond to stimuli. One of the regulatory patterns was a single auto-regulator that can self-regulate its expression or activity. Through analysis, nine auto-regulator TFs were identified (*Figure 2B*). This number likely represents a conservative estimate, as experiments may not optimally capture auto-regulatory events that depend on native expression levels or specific physiological conditions. Compared with non-auto-regulators, auto-regulators usually tend to be repressors, which play important roles in maintaining a steady state (*Alon, 2007*; *Burda et al., 2011*). For example, the key T3SS activators PsrA and HrpL can repress their own expression in *P. aeruginosa* and *P. savastanoi*, respectively (*Huang et al., 2022*; *Shen et al., 2006*; *Kojic et al., 2005*).

In addition, the most fundamental regulatory pattern was the ternary TF motif. We computed all possible motifs, which are listed in *Figure 2C* and *Supplementary file 4*. The basic triangular motifs were monodirectional regulatory structures with five different motifs. The most frequently observed motif (6,535 occurrences) was motif_2, wherein two different TFs co-regulate another TF. The remaining eight types of motifs, which contained toggle switches, occurred much less frequently than the basic motifs. The more complex the relationships among TFs, the lesser their occurrence. In particular, motif_10, motif_12, and motif_13, characterised by two or three mutually regulating TFs, were not found even once (*Figure 2C*). The high occurrence of different TFs in diverse motifs revealed a complex and flexible regulatory network in *P. aeruginosa*. Next, we constructed a basic motif_3 regulatory network based on potential TF-TF interactions to detect the regulatory preferences based on DBD types (*Figure 2D*). Basic motif_3 was a typical hierarchical regulatory network in which TF1

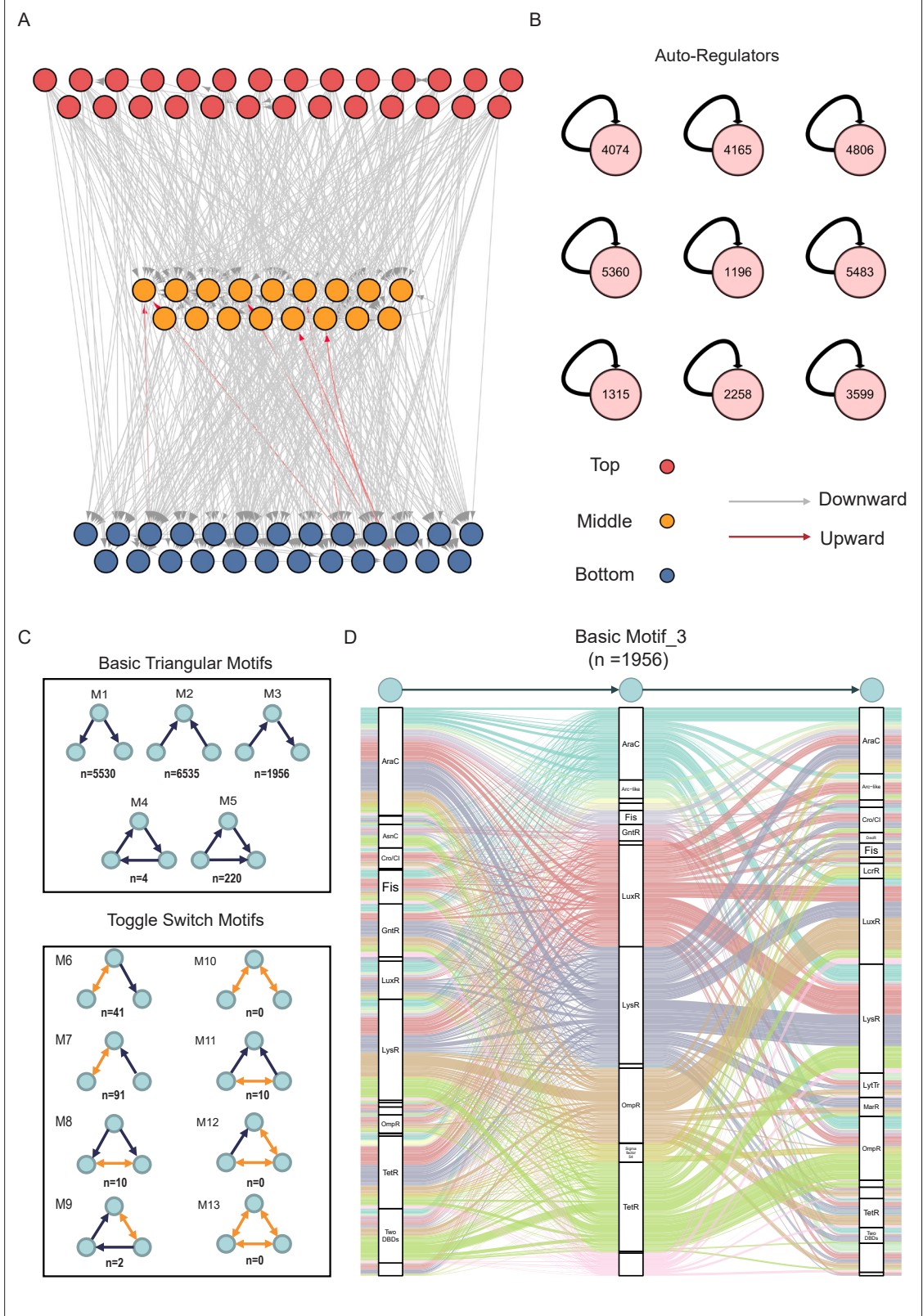

**Figure 2.** Hierarchical networks and network motifs. (**A**) Overview of hierarchy regulatory network after removing transcription factors (TFs) with degree $O + I <= 10$. Nodes indicate TFs, and the colour represents the hierarchical level. The top level was highlighted in red, the middle in yellow, and the bottom highlighted in blue. The edges with arrows indicate the regulatory direction. Grey means downward-pointing, and red means upward-pointing. (**B**) The auto-regulator motif of nine TFs. (**C**) Three TF motif occurrences with five basic triangular motifs and eight toggle switch motifs. The circle

*Figure 2 continued on next page*

*Figure 2 continued*

presents TF, and the arrow indicates the regulatory direction. (**D**) Alluvial diagram reveals basic triangle motif 3 (*n* = 1956) depending on DBD types. The colour of splines is highlighted in different DBD families, and the name of DBD families is labelled.

The online version of this article includes the following figure supplement(s) for figure 2:

**Figure supplement 1.** Distribution of hierarchy height *h*.

regulates TF2, which in turn regulates TF3. We found that the AraC, Fis, and GntR families tended to harbour top-level TFs rather than middle- and bottom-level TFs, while the OmpR family tended to harbour TFs from middle and bottom levels more than those from the top level. Furthermore, the LysR and LuxR families revealed non-significant differences among these three TF levels (*Figure 2D*). Taken together, our integrated hierarchical regulatory profiling revealed a multifaceted TF-TF connection, reflecting the regulatory preferences of TFs and DBD types in *P. aeruginosa*.

## Clustering of 103 TF-binding motifs

According to their binding sites, a total of 103 binding motifs were determined. The TFs were clustered based on the positional weight matrix similarity of their motifs. A circular phylogenetic tree was constructed using hierarchical clustering of the pairwise similarity matrix (*Figure 3—figure supplement 1A* and *Supplementary file 5*). Each node in the tree represents a TF, colour-highlighted according to its DBD family classification. The clustering analysis revealed four distinct clusters of TFs, one each coloured light green (cluster 1), light blue (cluster 2), purple (cluster 3), and pink (cluster 4). These clusters indicate groups of TFs with similar binding properties or regulatory functions. To validate the identified motifs, we compared them against existing motifs in the RegPrecise database (*Novichkov et al., 2013*), and we found the motif of PA3587 exhibited similarity to motifs of its orthologs in other *Pseudomonadaceae* species (*Figure 3—figure supplement 1B*).

Within each identified cluster, TFs tend to share conserved motif patterns, indicating potential functional relationships. For instance, it was found that the TFs in cluster 1 were enriched in adenine (A) and thymine (T) bases. Further, cluster 4 was more likely to have an inverted repeat (IR) sequence, such as PA0367 and PA1015. Clustering of TF-binding motifs identified groups of TFs with similar intrinsic DNA-binding specificities. As expected, many clusters contained TFs from the same DBD families, reflecting evolutionary conservation and potential functional redundancy or competitive binding at shared regulatory elements. Notably, the clustering also uncovered associations between TFs from different DBD families, suggesting convergent evolution of binding specificity or novel regulatory interactions that warrant further investigation.

## Genome-wide synergistic co-association of TFs in *P. aeruginosa*

To further investigate the crosstalk among TFs in *P. aeruginosa*, we computed the co-association scores based on the overlaps between the peaks of all pairs of TFs and ChIP-seq data (*Figure 3—figure supplement 2A* and *Supplementary file 6*). Among a total of 16,175 TF pairs, we found 5716 significant TF interactions with a co-association score of up to 0.1 based on an elbow statistic (*Figure 3—figure supplement 2B*). The relationships among TFs can be further determined as core clusters if their co-association scores are more than 0.4. The clustering analysis of core networks using Glay clustering (*Su et al., 2010*) resulted in seven associated modules, each containing a diverse number of TFs with the DBD families highlighted in different colours (*Figure 3A*).

Next, we summarised the potential functional crosstalk among the middle-scale co-associated five TFs (PA2417, PA2718, PA0756, PmiR, and AgtR) in cluster 3. Among a total of 1241 target genes, 372 were co-bound by at least three TFs, and 35 were co-bound by all five TFs, indicating that these TFs have a high level of collaboration in regulatory patterns (*Figure 3B*). There were smaller intersections among different combinations of TFs, highlighting potential co-regulation or shared binding sites. Binding targets co-bound by more than four TFs are summarised in *Figure 3C*. The dense network and connectivity indicate a complex regulatory landscape, where these TFs may orchestrate a coordinated regulation of gene expression. For example, peak visualisation of PA2504 revealed distinct binding patterns of these five TFs compared with the control group (*Figure 3D*). PA2504 is the sole partner of the PppH RNA hydrolase related to several important cellular factors (*Drabinska et al., 2021*). In addition, four of the five TFs (barring PA2718) can bind to the promoter region of *pqsH*,

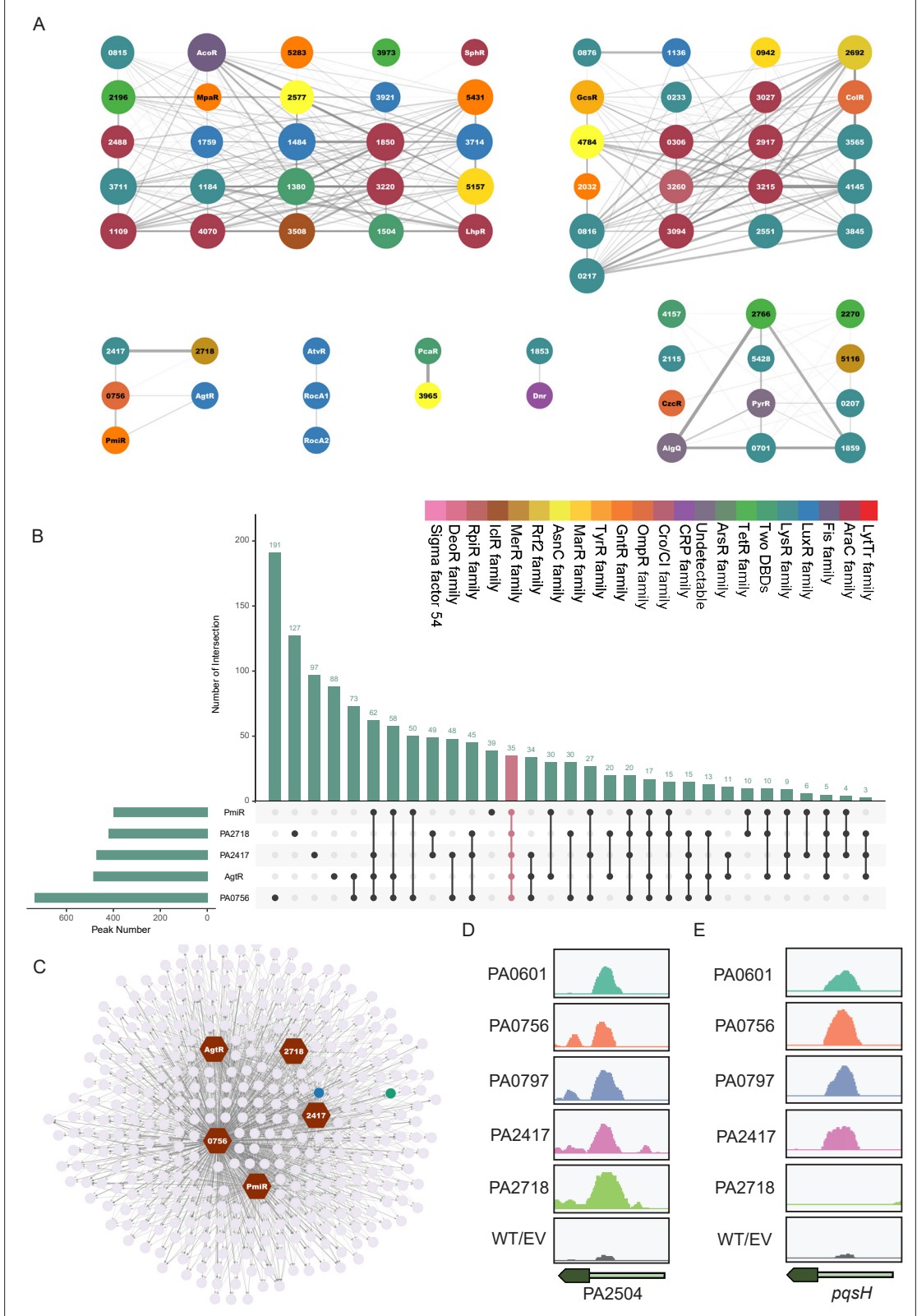

**Figure 3.** Core co-association regulatory networks. (**A**) Core clusters of significant co-binding patterns of TFs. Each TF is highlighted in different colours based on DBD types. The co-association score by pair of TFs was calculated by Jaccard statistics, which measures the ratio of the number of base pairs in overlapped binding peaks on both TFs to the number of base pairs in their union. (**B**) The histogram's overlapped target genes of TFs in cluster 3

*Figure 3 continued on next page*

*Figure 3 continued*

represent the number of target peaks in the individual/overlapped set. (**C**) The network of co-regulation of TFs in cluster 3 with co-bound targets of more than 4. (**D, E**). Genome browser view of TFs in cluster 3 binding intensities at the PA2504 and *pqsH* locus.

The online version of this article includes the following figure supplement(s) for figure 3:

**Figure supplement 1.** Circular phylogenetic tree of TF-binding motifs.

**Figure supplement 2.** Co-association network of TFs.

which is involved in the *pqs* pathway (**Figure 3E**). Notably, PmiR has previously been verified to bind to the *pqsH* promoter region and thereby regulate virulence (**Cui et al., 2022**). PA2504 and *pqsH* are highlighted in blue and green, respectively, in **Figure 3C**.

## Identified potential master regulators are associated with virulence

After elucidating the binding targets of 172 TFs, we investigated their potential functions in the virulence-related pathways and pathogenesis of *P. aeruginosa*. For this purpose, we performed an enrichment analysis using a hypergeometric test to define the master regulators (BH-adjusted *P* < 0.05) that might play important roles in regulating specific pathways. We primarily focused on TF-binding profiles involving the promoter regions of genes involved in nine pathways, namely QS, motility, biofilm production, antibiotic resistance, T6SS, T3SS, reactive oxygen species (ROS) resistance, pyocyanin production, and siderophore production. Accordingly, 134 of the 172 TFs were found to bind to at least one target gene associated with the abovementioned virulent pathways, and 24 were identified as master regulators of genes involved in six pathways, namely siderophore production, biofilm production, pyocyanin production, QS, motility, and ROS resistance pathways (**Figure 4A** and **Supplementary file 7**).

To experimentally validate the regulatory interactions identified by ChIP-seq, we performed biochemical and genetic analyses on selected TFs. First, we conducted Electrophoretic Mobility Shift Assays (EMSA) for four TFs, including PA0167, PA0815, PA1380, and PA3094, using DNA fragments containing their predicted binding sites from target gene promoters. These TFs showed specific binding to their cognate DNA sequences (**Figure 4—figure supplement 1A–D**), confirming the direct binding of the ChIP-seq-identified interactions.

To further validate the functional regulatory roles of these TFs, we constructed clean deletion mutants of PA1380 and PA3094 (ΔPA1380 and ΔPA3094) along with their complemented strains (ΔPA1380/p and ΔPA3094/p). Reverse transcription quantitative PCR (RT-qPCR) analysis revealed that PA1380 positively regulates the expression of *cupB1* and *cupB3* (**Figure 4—figure supplement 1E**), two genes within the CupB fimbrial cluster identified as ChIP-seq targets. Similarly, PA3094 was confirmed to positively regulate *lecA* expression (**Figure 4—figure supplement 1F**), which encodes a lectin involved in biofilm formation and host interactions (**Chemani et al., 2009**). Expression of these target genes was restored to wild-type (WT) levels in the complemented strains, validating the regulatory relationships predicted by ChIP-seq. These combined biochemical and genetic validations demonstrate the accuracy and biological relevance of our TF-binding data.

The intersection of master regulators across these virulent pathways revealed that six TFs (PA0167, PyrR, PA0707, PA0877, PA1504, and PA1484) only played an important role in regulating the motility pathway, and four TFs (PA1380, PA0815, PA5428, and PA3973) jointly regulated the motility and other pathways, including biofilm production, pyocyanin production, and QS pathways (**Figure 4B**). Specifically, PA0815 co-regulated motility, biofilm production, and QS pathways. PA1380 jointly regulated motility and pyocyanin production pathways. PA3973 and PA5428 jointly regulated motility and biofilm production pathways. The regulatory network of these four master regulators and their target genes is summarised in **Figure 4—figure supplement 1G**, represented by lines connecting the TFs to the targets involved in specific pathways.

## Predicted functions of regulators characterised in two metabolic pathways

*P. aeruginosa* uses different global regulators associated with metabolism that ensure its survival and enhance its adaptability under fluctuating environmental conditions. For instance, it uses different regulatory mechanisms to respond to nutrient changes, oxygen limitation, or CF-associated lung

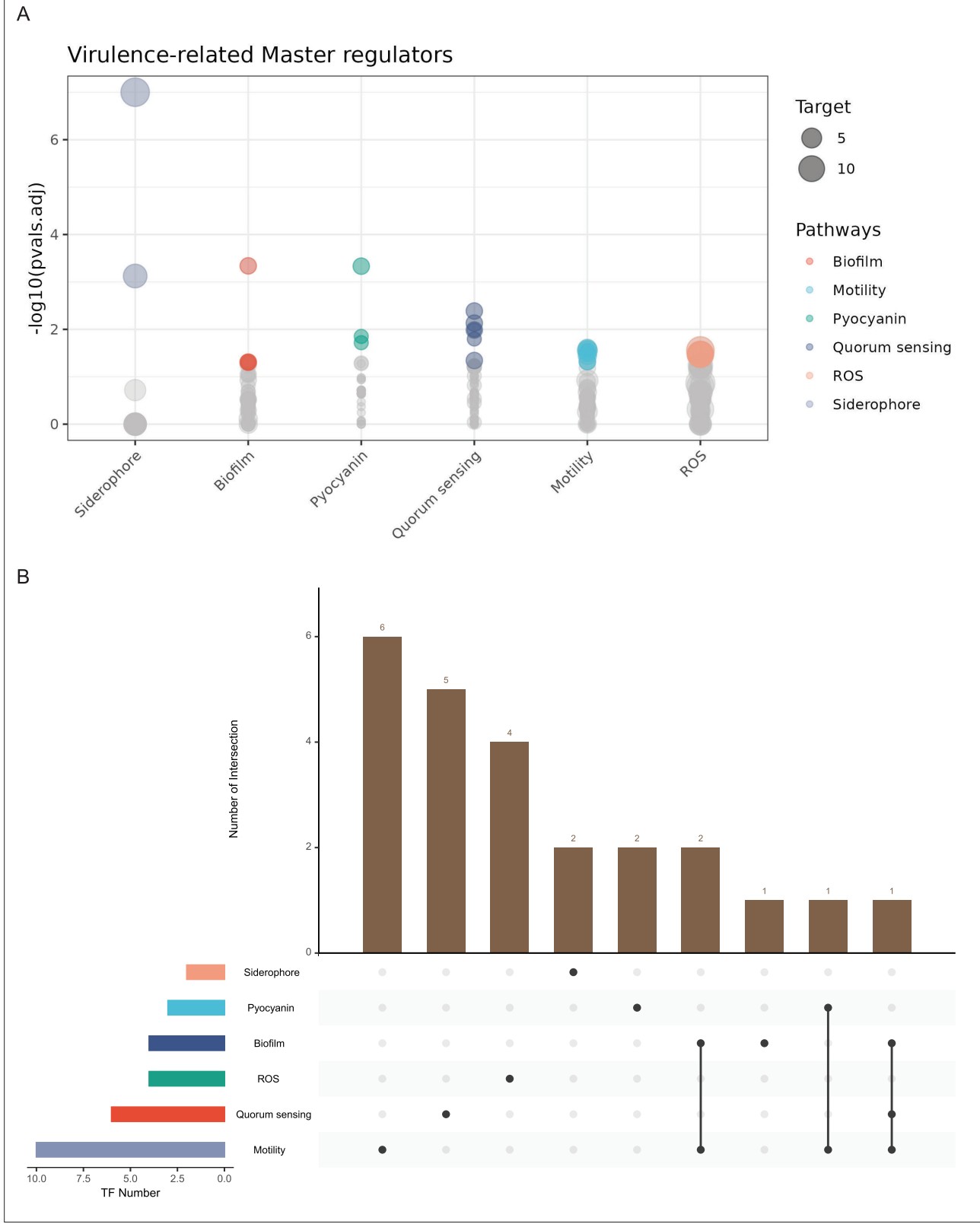

**Figure 4.** Newly identified virulence-related master regulators. (**A**) Overview of all identified master regulators related to six pathways, including QS, motility, biofilm, siderophore, pyocyanin, and ROS. Each circle represents one transcription factor (TF), with the height indicating the significance, quantified by −log$_{10}$ (p.adjust), and the size indicates the number of targets associated with the virulence pathway. (**B**) Intersection of master regulators

*Figure 4 continued on next page*

*Figure 4 continued*

in six virulence pathways. The bar chart on the top shows the number of intersections, while the matrix below indicates which TFs are involved in specific biological processes such as siderophore production, pyocyanin production, biofilm formation, ROS response, QS, and motility.

The online version of this article includes the following figure supplement(s) for figure 4:

**Figure supplement 1.** Validation and co-regulation of virulence-related master regulators.

**Figure supplement 2.** Regulators involved in tricarboxylic acid (TCA) and ribosome pathways.

infection conditions (*Moreno and Rojo, 2023*; *Dolan et al., 2020*; *Oberhardt et al., 2010*; *Korgaonkar and Whiteley, 2011*).

To define the core TFs that regulate metabolism, we performed master regulator analysis in metabolic pathways (p.adjust < 0.05) based on the TFBSs located in the promoter regions revealed by ChIP-seq for all TFs. We identified 14 TFs potentially involved in two pathways, namely the tricarboxylic acid (TCA) cycle and ribosome function pathways (*Figure 4—figure supplement 2A, B*). The radial plot highlights the master regulators within the TCA cycle and ribosome function pathways, and their joint regulation with their targets is presented in the regulatory network below. PA2957 and PA5438 were found to play putative roles in the TCA cycle by potentially binding to the promoter of certain genes, including *icd*, *sucA*, and *sdhC*, indicating that they coordinate the expression of genes crucial for central metabolism and energy production (*Figure 4—figure supplement 1B*). Furthermore, certain TFs, such as PA0611, PA0403 (PyrR), and PA1283, were found to be involved in regulating a broad array of ribosomal genes, such as *rpsB*, *rpsA*, and *rplL*. Notably, TF PyrR was found to play a putative role in regulating motility in *P. aeruginosa*. Overall, these regulators, which are related to the core metabolic pathways that enhance the adaptation of *P. aeruginosa* to different infection conditions and contribute to its pathogenicity, could offer potential drug targets for tackling *P. aeruginosa* infection in the future.

## A global summary of transcription regulatory networks and functions in *P. aeruginosa*

After studying the TFBSs of most of the TFs in *P. aeruginosa* both *in vivo* and *in vitro*, we organised these datasets to yield a comprehensive connection of virulence-related master regulators. The QS and motility pathways consisting of 18 and 20 TFs, respectively, were the key pathways in the virulent landscape and showed a close collaboration with the other five pathways (*Figure 5A*). Four TFs (ExsA, CprR, AlpR, and PA2449) were involved in T3SS, a direct host infection mechanism used by *P. aeruginosa*. Additionally, six TFs (PA4436, PA0929, PA3771, PA2534, PA3699, and PA5218) were involved in siderophore production, as identified using HT-SELEX, and showed correlations with only one TF, PA2534, involved in the ROS resistance pathway. Four TFs (CprR, PA0708, PA1520, and PA2479) were characterised in the stringent response and persister cell pathways and were correlated with other virulence-related pathways, namely QS, ROS resistance, T3SS, and antibiotic resistance pathways.

To investigate the phylogenetic classification of the binding targets of the identified TFs on a genome-wide scale, we derived a global atlas revealing the relationships between the protein functions of target genes and the family categories of TFs (*Figure 5B*). The Clusters of Orthologous Groups of proteins (COGs) database was used to classify proteins based on the orthology concept (*Tatusov et al., 2000*). Based on the annotated COGs of *P. aeruginosa*, we found that most of the TF targets were involved in metabolism, followed by poorly characterised categories. The *P. aeruginosa* PAO1 genome is 6.3 Mbp long, and approximately 40% of it remains uncharacterised but is believed to encode well-conserved hypothetical proteins (HPs), which might have indispensable and similar functions. Given that most of the TF targets are poorly characterised, it is crucial to study these HPs in the future to provide more insights into *P. aeruginosa* pathogenicity. Altogether, our results provide a resource of TFs and their putative target genes in *P. aeruginosa* and, consequently, a basis for understanding the regulatory network of virulence and orthological functions.

## A web-based database of all TFBSs in *P. aeruginosa*

To enable a quick search of the TFBSs in *P. aeruginosa*, we developed a web-based database containing our ChIP-seq and HT-SELEX results at https://jiadhuang0417.shinyapps.io/PATF_Net/, which is available to the public. The homepage briefly describes the background information and schematic

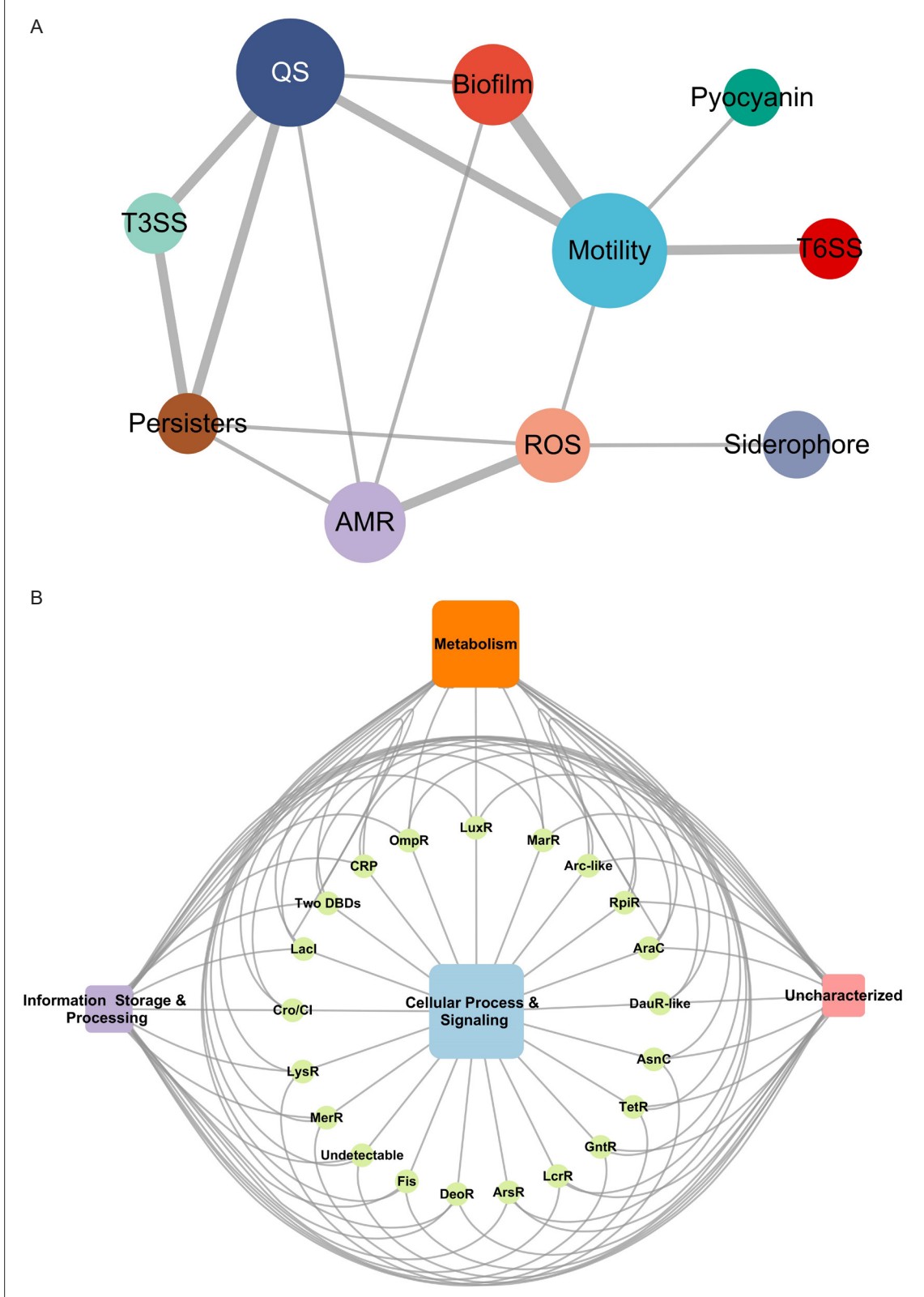

**Figure 5.** Overview of transcriptional network of TFs in *P. aeruginosa*. (**A**) The interaction network of virulence-related master regulators in *P. aeruginosa*. The 10 virulence pathways are highlighted in different colours. The size of the circle represents the degree, and the width of the edges indicates the overlapped number of TFs between two pathways. (**B**) The target annotated using the COG database shows four orthology classification profiling from different DBD types TFs. The size of the rectangle indicates the number of targets.

workflow of ChIP-seq and HT-SELEX. The data page provides two types of data: a network plot of TFs and targets, and detailed TF–target information in a table format. The network plot illustrates the top 50 peaks for ease of visualisation. The code was generated using R, and the figures were created using BioRender (ChIP-seq: https://biorender.com/zru6mz7; HT-SELEX: https://biorender.com/2mwef9b).

The database offers multiple search modalities to facilitate data exploration: users can perform TF-centric searches to query binding sites, target genes, and regulatory networks for individual TFs, or utilise the target gene search function to identify all TFs that regulate any gene of interest by entering its locus tag. To connect regulatory data with biological function, we have implemented a virulence pathway browser that allows users to explore TF-binding patterns across curated gene sets for major *P. aeruginosa* virulence pathways. Interactive visualisation tools, including network graphs and binding profile plots, facilitate intuitive exploration of regulatory relationships. The primary purpose of PATF_Net is to store, search, and mine valuable information on *P. aeruginosa* TFs for researchers investigating *P. aeruginosa* infection. The current resource is based on the reference strain PAO1, which serves as the foundation for most *P. aeruginosa* molecular studies and allows direct integration with existing genomic annotations and functional data. However, *P. aeruginosa* exhibits substantial genomic diversity across clinical isolates, and strain-specific differences in TF-binding patterns may contribute to phenotypic variation in virulence, antibiotic resistance, and host adaptation. Extension of this resource to include strain-specific regulatory maps from diverse clinical isolates would provide valuable insights into the regulatory basis and represent an important direction for future investigation.

## Conservation and evolution of TFs in *P. aeruginosa*

Differences between species are mainly due to differences in genes, whereas essential genes are usually well conserved, especially in subspecies. To investigate the conservation of TFs in *P. aeruginosa*, we performed a pan-genome analysis of 100 strains containing model strains and clinical strains using the Roary software (*Page et al., 2015*). A total of 21,432 genes were identified across the 100 *P. aeruginosa* strains, of which only 4544 genes were classified as core genes (existing in more than 99% of strains), revealing that most of the genes were cloud genes (existing in less than 15% of strains) (*Figure 6A*). However, when focusing on the TFs among these 100 strains, almost 89% of TFs (331 of 373) were found to be well-conserved (*Figure 6B*), which implied that these recognised TFs from PAO1 play essential roles in biological activities in all of the *P. aeruginosa* subspecies.

The bar plot provides detailed information on the rest of the non-core TFs in PAO1, in addition to several important TFs, such as LasR and ExsA, that are lacking in some strains (*Figure 6C*). As is known, the *lasR* mutant of *P. aeruginosa* is associated with CF lung disease progression and is considered a marker of an early CF-adaptive phenotype (*Hoffman et al., 2009*). ExsA is considered an important activator of the T3SS pathway in *P. aeruginosa*, which is characterised by an acute infection phenotype. Additionally, VqsM might have a relatively special function in PAO1 among *P. aeruginosa* subspecies in addition to playing an important role in the QS and antibiotic resistance pathways. Taken together, the core- and non-core TFs were both vital and worth studying for understanding the metabolism and virulence-related regulatory mechanisms in *P. aeruginosa*.

To further investigate the conservation of TFs in *P. aeruginosa*, we re-analysed the ChIP-seq data of PhoB and RpoN in the PA14 strain (*Bielecki et al., 2015*; *Allsopp et al., 2022*) and compared them with those in the PAO1 strain. *Figure 6—figure supplement 1A* shows genome-wide binding peaks of PhoB and RpoN throughout the genomes of PAO1 and PA14. We found that in PA14 and PAO1, PhoB had 725 and 63 peaks, while RpoN yielded 363 and 1238 peaks, respectively. Next, we used MEME (*Bailey et al., 2009*) to compare the motifs of PhoB and RpoN in PAO1 and PA14 based on their binding sequences from ChIP-seq (*Figure 6—figure supplement 1B*). PhoB and RpoN shared a well-conserved motif in the two strains even though they showed different peak distributions in the genomes. They had different numbers of peaks, and of the 690 and 345 respective annotation targets of PhoB and RpoN in PA14, 114 and 66 targets, respectively, were unique to PA14. The rest of the targets were homologous between PAO1 and PA14. We next performed target annotation and observed that the genes regulated by these two TFs were significantly (Fisher's test) overrepresented in the two tested strains (*Figure 6—figure supplement 1C*). For example, RpoN has been found to play an important role in regulating T6SS in both PAO1 and PA14 (*Allsopp et al., 2022*; *Shao et al., 2018*). Overall, the conservation of the representative TFs PhoB and RpoN in PAO1 and PA14 indicates that conserved TFs may share similar functions in

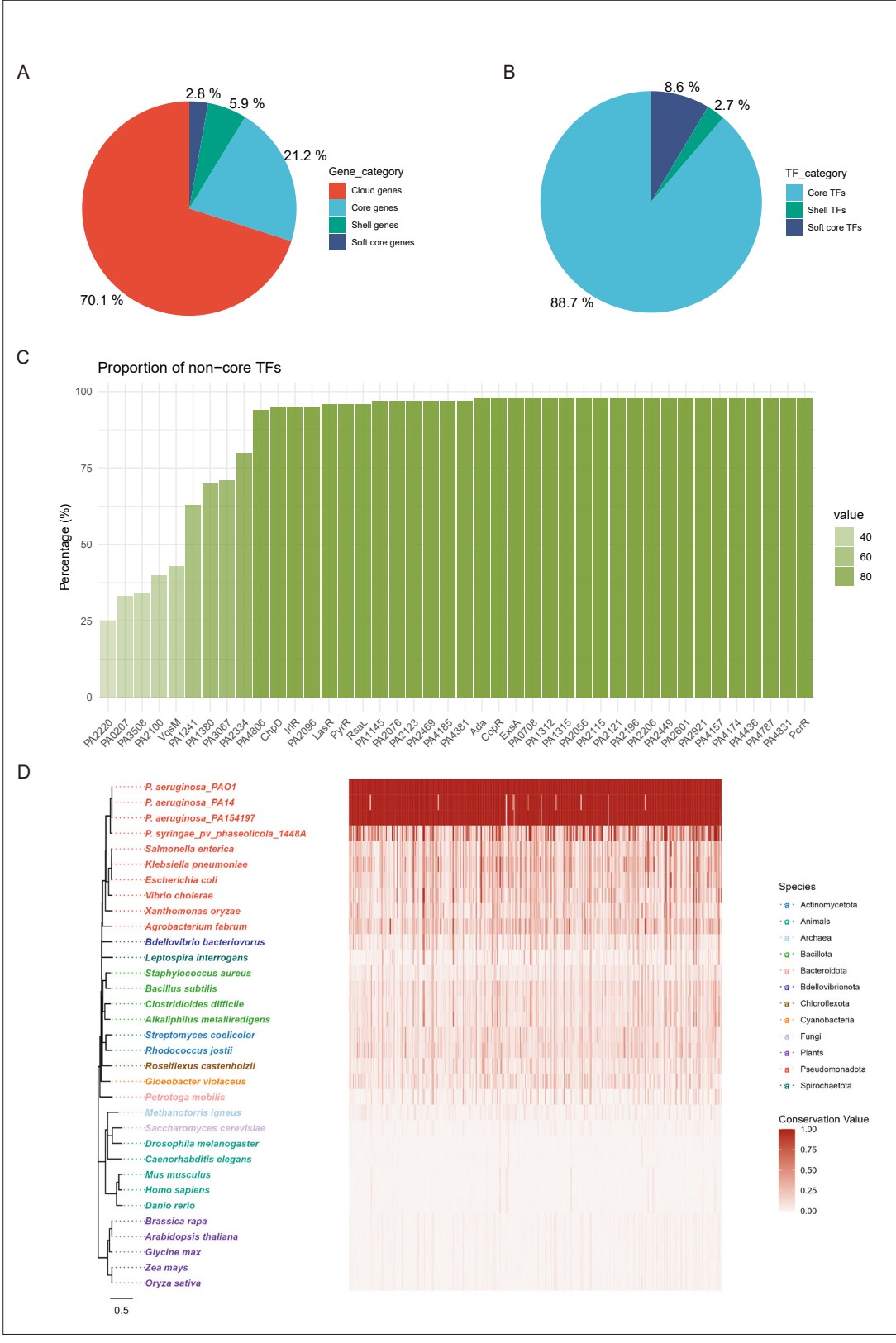

**Figure 6.** Conservation and variability of TFs in PAO1. (**A**) The pie chart shows the proportions of genes categorised by their presence across *P. aeruginosa* strains for all genes. (**B**) The pie chart shows the distribution of TFs identified from PAO1 across different conservation categories. (**C**) The bar plot of the proportion for non-core TFs. Genes are categorised based on their presence frequency across *P. aeruginosa* strains: Core genes (present

*Figure 6 continued on next page*

*Figure 6 continued*

in 99–100% strains), Soft core genes (present in 95–99% strains), Shell genes (present in 15–95% strains), and Cloud genes (present in 0–15% strains). (**D**) The conservation and evolutionary relationship of all 373 TFs in PAO1 among bacteria, archaebacteria, fungi, plants, and animals. The conservation value was normalised after blastp alignment (*Coordinators, 2013*). The phylogenetic trees were constructed using MEGA11 (*Tamura et al., 2021*) and plotted via R package ggtree (*Yu et al., 2017*).

The online version of this article includes the following figure supplement(s) for figure 6:

**Figure supplement 1.** Conserved TFs in *P. aeruginosa*.

different species. This knowledge is expected to guide drug development for different infectious strains.

TFs play a pivotal role in regulating gene expression, and studying their conservation and evolution could provide insights into how gene expression patterns are conserved or modified across different species and help to predict gene regulatory networks in different species. For example, the conservation of a TF across multiple species suggests that its targets and regulatory interactions are likely to be conserved as well. Therefore, to study the conservation and evolution of *P. aeruginosa* TFs across other species, we conducted phylogenetic analyses of the 373 TFs of *P. aeruginosa* across other bacteria, fungi, archaea, plants, and animals. We found that three *P. aeruginosa* species showed the most conserved TFs, followed by *P. savastanoi*, a phytopathogen (*Figure 6D*). Compared with bacteria, the conservation values of the TFs were lower in fungi, archaea, plants, and animals. However, several orthologues of TFs were still noted among these organisms. For instance, PA2032 had more than 25% and approximately 30% identity with aminoadipate aminotransferase in humans and mice, respectively. The phylogenetic tree of PA2032 across bacteria, archaea, fungi, plants, and animals, with PAO1 as the root, revealed that the bacterial TFs (purple) indicate a high degree of conservation within prokaryotes, suggesting a fundamental role in core regulatory processes (*Figure 6—figure supplement 1D*). In contrast, eukaryotic TFs (fungi, plants, and animals) form distinct clades with longer branch lengths, indicating significant divergence and specialisation during eukaryotic evolution. These findings suggest that while TF is conserved across domains of life, its functional roles and regulatory mechanisms have undergone substantial diversification in eukaryotes. Altogether, the high-level evolutionary conservation of TFs in *P. aeruginosa* strongly suggests that the regulatory mechanisms are common to a wide range of bacteria.

## Discussion

Bacterial pathogens use many mechanisms to establish infection and cause diseases in human hosts, making them a major public health concern worldwide. These bacteria can secrete a wide range of molecular particles that recognise and bind to host cell targets, thus damaging or preventing host responses (*Wilson et al., 2002*). Pathogenic bacteria use strategies to adapt to environmental stressors such as nutrient limitation or antibiotic treatment (*Blair et al., 2015*). The underlying mechanism of these strategies involves sensing external signals and then responding to different environmental conditions. TFs are key molecules involved in responding to host or environmental signals by precisely modulating transcription levels; hence, their pivotal roles in infection conditions cannot be overstated. Nevertheless, our comprehension of bacterial TFs remains considerably limited, even in extensively examined model organisms such as *P. aeruginosa*. To address this knowledge gap and shed more light on TFs in *P. aeruginosa*, we conducted ChIP-seq experiments aimed at elucidating the TFBSs for 172 TFs with relatively unexplored biological functions.

ChIP-seq is a valuable technique to obtain high-resolution DNA-protein interaction mapping on a genome-wide scale to study the potential binding targets of TFs. The technique is highly sensitive and can detect low-abundance DNA-binding events, allowing the detection of weak TF-DNA interactions. To enhance the visualisation of more subtle binding peaks, particularly when investigating co-associations and TFs lacking well-defined signals, we devised a strategy involving TF-overexpressing plasmids. *In vitro* assays, such as DAP-seq or HT-SELEX, offer superior scalability to ChIP-seq but present their own challenges (*Trouillon et al., 2021*; *Baumgart et al., 2021*). Our analysis revealed that TF-binding events occur within coding regions, which is consistent with a previous study demonstrating that bacterial TFs possess binding capabilities for coding regions and can regulate transcription through

multiple mechanisms (*Hua et al., 2022*). Besides, it may also regulate RNA stability, warranting integration with translatomics or epigenomics data. These findings extend the basic function of TFs to recognise specific DNA sequences (i.e., motifs) at the promoter region and then upregulate or downregulate the transcription of the target gene (*Perez-Rueda et al., 2018*). Notably, the literature underscores the collaborative nature of TFs, which often exhibit interactions with other auxiliary proteins at multiple promoter sites (*Browning et al., 2019*). As shown in *Figure 3* and *Figure 3—figure supplement 1*, we present a TF-binding motifs clustering tree and a TF co-association regulatory network based on ChIP-seq data and core clusters of TFs with high co-association scores. Overall, our above-described analysis has illustrated a regulatory atlas to study the biological functions of *P. aeruginosa* TFs in specific contexts.

However, several limitations of the ChIP-seq approach should be acknowledged. Firstly, TF overexpression ensures sufficient protein levels for ChIP-seq signal detection but does not guarantee that all TFs are in their active conformational states, as many bacterial TFs require allosteric activation by metabolites, cofactors, or post-translational modifications. The cells under standard laboratory conditions, which may not activate all TFs to their maximal regulatory states, potentially leading to underestimation of condition-specific binding peaks. Secondly, while we observed TF binding at thousands of genomic sites, binding per se does not equate to functional regulation, as chromatin context, cofactor availability, and competitive binding all influence regulatory outcomes.

*P. aeruginosa* uses TFs to control biological functions, including metabolism and virulence. Given the public health burden of *P. aeruginosa* infection, many studies have investigated the roles of TFs in this pathogen. These studies have characterised the regulatory mechanisms of individual TFs, such as FleQ, AlgR, LasR, VqsM, and PvrA (*Liang et al., 2014*; *Arora et al., 1997*; *Kong et al., 2015*; *Gilbert et al., 2009*; *Pan et al., 2020*). Additionally, a TRN comprising 690 genes and 1020 regulatory interactions with six biological modules and main motifs was established in 2011 using published data (*Galán-Vásquez et al., 2011*). Furthermore, we developed a PAGnet containing 20 key virulence-associated TFs based on ChIP-seq and RNA-seq data (*Huang et al., 2019*). A previous study used machine learning to identify modules of a group of genes related to known TFs from published data (*Rajput et al., 2022*). Another study used DAP-seq to determine a regulatory network of 55 RRs in *P. aeruginosa* (*Trouillon et al., 2021*). To further identify and characterise unknown TFs, we used HT-SELEX to describe the binding specificities of 182 TFs (*Wang et al., 2021*). Compared with the abovementioned studies, the present work profiled a comprehensive regulatory network of *P. aeruginosa*, combining existing and new data with experimental verification.

Additionally, although the TRN analysis revealed organisational patterns in *P. aeruginosa* regulatory network, the functional significance of these topological features, including their specific contributions to pathogenicity, metabolic adaptation, and antibiotic resistance, remains to be experimentally determined in the future work. The hierarchical structure and regulatory motifs we identified represent objective network properties derived from our binding data, but translating these structural observations into mechanistic understanding will require condition-specific functional studies, genetic validation, and phenotypic characterisation. Our analysis provided a systematic framework and generated testable hypotheses rather than definitive functional conclusions. Nevertheless, these network-level organisational principles provided value to the community as a foundational reference, similar to other regulatory network maps (*Galán-Vásquez et al., 2011*) that were useful even before comprehensive validation.

We confirmed the functions and provided potential molecular mechanisms of a number of previously identified TFs to explain their phenotypes. PA0797 is known to regulate the *pqs* system and pyocyanin production (*Cui et al., 2022*). In the present study, it was also found to bind to the *pqsH* promoter region and its motif was visualised. PA5428 was found to bind to the promoter regions of *aceA* and *glcB* genes (*Hwang et al., 2021*), which was also demonstrated in our ChIP-seq results. PA4381 (CloR) was found to be associated with polymyxin resistance in a previous study (*Gutu et al., 2013*) and to be possibly related to ROS resistance in the present study. Furthermore, PA5032 plays a putative role in biofilm regulation and also forms an operon with PA5033, an HP associated with biofilm formation (*Zhang et al., 2013*).

While our phylogenetic analysis reveals varying degrees of TF conservation across bacterial species, the functional implications of this conservation remain to be fully explored. Many *P. aeruginosa* TFs have clear orthologs in both Gram-negative (e.g., *Klebsiella pneumoniae*) and Gram-positive

pathogens (e.g., *Bacillus cereus*), yet whether these orthologs regulate similar target genes and biological processes is largely unknown. Future comparative profiling of orthologous TFs could reveal the extent to which regulatory network architecture is conserved versus rewired during bacterial evolution, potentially identifying core regulatory modules governing universal bacterial strategies versus species-specific innovations. Such cross-species comparisons would enhance our understanding of regulatory network evolution and enable functional prediction in less well-characterised pathogens based on homology to experimentally validated *P. aeruginosa* regulators.

Our study is the first to illustrate the hierarchical regulatory network and ternary motifs of and genome-wide co-association relationships among TFs in *P. aeruginosa*, contributing to a better understanding of the fundamental traits of pathogenicity of this species. Traditional bacterial eradication strategies involving antibiotics often focus on pathogen extermination, yet this approach can inadvertently foster selective growth advantages and precipitate the emergence of drug-resistant strains. Therefore, prioritising TF-targeted drugs could potentially alleviate selective growth pressure and enhance future preventative measures against *P. aeruginosa* infection. The extensive datasets generated in this study offer valuable insights into understanding and targeting *P. aeruginosa* pathogenicity. The genome-wide binding profiles can be systematically analysed through our hierarchical regulatory network framework to decode complex virulence mechanisms. The virulence-related master regulators and core regulatory clusters identified in this study highlighted key nodes of transcriptional control. Understanding these regulatory relationships is particularly valuable for identifying targets whose modulation would significantly impact virulence while accounting for potential compensatory mechanisms. This knowledge base thus provides a foundation for developing targeted approaches to combat *P. aeruginosa* infections, moving beyond traditional antibiotic strategies towards more sophisticated interventions based on regulatory network manipulation.

## Methods
### Strains, plasmids, and primers
The bacterial strains, plasmids, and primers used in the present study are listed in *Supplementary file 8*. The *P. aeruginosa* PAO1 strain and its derivatives were grown at 37°C in LB (Luria-Bertani) broth with shaking at 220 rpm or on LB agar plates. Antibiotics were used for *E. coli* at the following concentrations: kanamycin at 50 μg/ ml and carbenicillin at 60 μg/ml.

### ChIP-seq and analysis
The chromatin immunoprecipitation (ChIP) procedures were modified from a previous study (*Blasco et al., 2012*). Briefly, for the VSVG-tagged, the ORF was amplified by PCR from the PAO1 genome and cloned into pAK1900 plasmid by HindIII/BamHI for the overexpressed TFs through HindIII site by using ClonExpress MultiS One Step Cloning Kit (Vazyme, China). WT *P. aeruginosa* containing empty pAK1900 or pAK1900-TF-VSV-G was cultured in LB medium supplemented at 37°C with shaking until mid-log phase ($OD_{600}$ = 0.6), then we cross-linked the samples with 1% formaldehyde for 10 min. Subsequently, cross-linking was stopped by the addition of 125 mM glycine. Samples were centrifuged and washed thrice with a Tris buffer (20 mM Tris-HCl, pH 7.5, 150 mM NaCl). The resulting pellets were resuspended in 500 μl IP buffer 50 mM HEPES–KOH [pH 7.5], 150 mM NaCl, 1 mM EDTA, 1% Triton X-100, 0.1% sodium deoxycholate, 0.1% SDS, and mini-protease inhibitor cocktail (Roche), and then the DNA was broken into pieces (100–300 bp) with an ultrasonic processor. Insoluble cellular debris was removed by centrifugation at 4°C, and the supernatant was used as the input sample in IP experiments. We next added 25 μl agarose-conjugated anti-VSV antibodies (Sigma) in the IP buffer. Washing, reverse cross-linked, and purification of the ChIP DNA were conducted. We used the NEXTflex ChIP-seq kit (Bio Scientific) to construct the DNA fragment library, and agarose gel was used to cut DNA fragments between 150 and 250 bp. After sequencing, the raw reads were trimmed by Trim Galore (*Felix Krueger et al., 2023*) with default parameters. Then, the filtered reads were aligned to the PAO1 genome (NC_002516) using bowtie2. Only the unique reads after alignment will be used to perform peak calling with MACS2 (p < 0.001). MEME-ChIP was used to identify consensus motifs with all peaks. TF target genes were then annotated by the R package ChIPpeakAnno.

## Identification of virulence-related master regulators

We adopted the definition of 'master regulator' from developmental biology (*Chan and Kyba, 2013*), where it refers to TFs that control the expression of multiple downstream genes governing a specific biological process or lineage commitment. In our context, we use 'master regulator' to designate TFs that coordinately regulate multiple genes within specific virulence-related pathways based on statistical enrichment criteria as described by *Fan et al., 2020*. Briefly, we first generated gene lists associated with nine pathways (*Shao et al., 2023*), including QS, motility, biofilm production, antibiotic resistance, T6SS, T3SS, ROS resistance, pyocyanin, and siderophores. These gene lists are provided in full detail at our PA_TFNet database. Then, we calculated the number of overlap genes between target genes of TF and genes involved in the above eight pathways. The statistical significance of the master regulator was identified using the Hypergeometric test (BH-adjusted $p < 0.05$). R package ggplot2 (*Wickham, 2016*) was used to visualise the result.

## GO analysis

GO enrichment analysis of TFs target genes was identified using R package clusterProfile (*Yu et al., 2012*). The enriched GO terms with BH-adjusted $p < 0.05$ were defined as significantly enriched.

## Pan-genome analysis

The information on the model and clinical *P. aeruginosa* strains (*Hu et al., 2019*; *Hu et al., 2021*) can be found in *Supplementary file 9*. Briefly, the genome sequence of all strains was annotated by prokka (*Seemann, 2014*) and the outputs were then used as input files for roary.

## Electrophoretic Mobility Shift Assay

DNA probes corresponding to predicted TF-binding regions were PCR-amplified from PAO1 genomic DNA. For binding reactions, 20 ng of purified DNA probe was incubated with varying concentrations of recombinant TF protein in 20 µl binding buffer containing 10 mM Tris-HCl (pH 7.4), 50 mM KCl, 5 mM $MgCl_2$, and 10% (vol/vol) glycerol. Reactions were incubated at room temperature for 20 min to allow protein–DNA complex formation. Samples were then resolved on 6% native polyacrylamide gels by electrophoresis at 90 V for 90 min in TBE buffer. Following electrophoresis, gels were stained with GelRed for 5 min and visualised using a ChemiDoc imaging system (Bio-Rad).

## Reverse transcription quantitative PCR

Bacterial cultures were grown in LB medium to mid-log phase ($OD_{600}$ = 0.6), and total RNA was extracted using the Bacterial Total RNA Isolation Kit (Sangon Biotech). RNA concentration and purity were assessed using a NanoDrop 2000 spectrophotometer (Thermo Fisher Scientific). cDNA synthesis was performed using 600 ng of total RNA with HiScript III RT SuperMix (Vazyme). Quantitative PCR reactions were carried out using ChamQ Universal SYBR qPCR Master Mix (Vazyme) on a QuantStudio Real-Time PCR System (Applied Biosystems). Relative gene expression levels were calculated using the $2^{-\Delta\Delta Ct}$ method, with expression in WT strain set as 1.0. All RT-qPCR experiments were performed with three biological replicates, each with three technical replicates.

## Acknowledgements

This study was funded by the Guangdong Major Project of Basic and Applied Basic Research (2020B0301030005), Health and Medical Research Fund (20190942), Shenzhen Science and Technology Fund (JCYJ20210324134000002), National Natural Science Foundation of China grants (32172358), Hong Kong Research Grants Council Collaborative Research Fund (CRF C7033-20G), General Research Funds of Hong Kong (11103221, 11101722, and 11102223), and Theme-based Research Scheme (T11-104/22-R). The funders had no role in study design, data collection, interpretation, or the decision to submit the work for publication.

## Additional information

### Funding

| Funder | Grant reference number | Author |
| --- | --- | --- |
| Guangdong Major Project of Basic and Applied Basic Research | 2020B0301030005 | Xin Deng |
| Shenzhen Science and Technology Fund | JCYJ20210324134000002 | Xin Deng |
| General Research Funds of Hong Kong | 11102223 | Xin Deng |
| Health and Medical Research Fund | 20190942 | Xin Deng |
| General Research Funds of Hong Kong | 11101722 | Xin Deng |
| General Research Funds of Hong Kong | 11103221 | Xin Deng |
| National Natural Science Foundation of China | 32172358 | Xin Deng |
| Hong Kong Research Grants Council Collaborative Research Fund | CRF C7033-20G | Xin Deng |
| Theme-based Research Scheme | T11-104/22-R | Xin Deng |

The funders had no role in study design, data collection, and interpretation, or the decision to submit the work for publication.

### Author contributions

Jiadai Huang, Conceptualization, Software, Formal analysis, Supervision, Validation, Investigation, Visualization, Methodology, Writing – original draft, Project administration, Writing – review and editing; Yue Sun, Resources, Validation, Investigation, Methodology, Writing – review and editing; Fang Chen, Validation, Investigation, Methodology, Writing – review and editing; Shumin Li, Beifang Lu, Visualization, Methodology; Xiangkai You, Zhe He, Validation, Methodology; Liangliang Han, Jingwei Li, Chunyan Yao, Tianmin Li, Methodology; Canfeng Hua, Supervision, Methodology; Yung-Fu Chang, Supervision, Writing – review and editing; Xin Deng, Conceptualization, Data curation, Supervision, Funding acquisition, Methodology, Project administration, Writing – review and editing

### Author ORCIDs

Jiadai Huang ⓘ https://orcid.org/0000-0002-4104-6836
Yue Sun ⓘ https://orcid.org/0009-0008-2843-0017
Fang Chen ⓘ https://orcid.org/0000-0002-2154-4440
Yung-Fu Chang ⓘ https://orcid.org/0000-0001-8902-3089
Xin Deng ⓘ https://orcid.org/0000-0003-1580-0089

Reviewer #1 (Public review): https://doi.org/10.7554/eLife.103346.4.sa1
Reviewer #3 (Public review): https://doi.org/10.7554/eLife.103346.4.sa2
Author response https://doi.org/10.7554/eLife.103346.4.sa3

## Additional files

### Supplementary files

Supplementary file 1. List of 172 ChIPed transcription factors (TFs).
Supplementary file 2. Enriched Gene Ontology (GO) terms of 172 ChIPed transcription factors (TFs).

Supplementary file 3. Hierarchical regulatory network.

Supplementary file 4. Ternary regulatory motifs.

Supplementary file 5. PWM pairwise similarity scores.

Supplementary file 6. Co-association network.

Supplementary file 7. Virulence-related master regulators.

Supplementary file 8. Strains and primers used in this study.

Supplementary file 9. Reference list of strains for pan-genome analysis.

MDAR checklist

## Data availability

The ChIP-seq data was uploaded to National Center for Biotechnology Information Gene Expression Omnibus database under accession GSE241603 and GSE271817. Analysis codes have been deposited at https://github.com/dengxinb2315/PS-PATRnet-code (copy archived at *Deng, 2026*).

The following datasets were generated:

| Author(s) | Year | Dataset title | Dataset URL | Database and Identifier |
|---|---|---|---|---|
| Deng X | 2025 | Global transcription factors analyses reveal hierarchy and synergism of regulatory networks and master virulence regulators in *Pseudomonas aeruginosa* | https://www.ncbi.nlm.nih.gov/geo/query/acc.cgi?acc=GSE241603 | NCBI Gene Expression Omnibus, GSE241603 |
| Deng X | 2025 | Global transcription factors analyses reveal hierarchy and synergism of regulatory networks and master virulence regulators in *Pseudomonas aeruginosa* 2 | https://www.ncbi.nlm.nih.gov/geo/query/acc.cgi?acc=GSE271817 | NCBI Gene Expression Omnibus, GSE271817 |

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
