## [Editor Report · eLife Assessment]

This study provides an **important**, comprehensive, large-scale dataset on transcription factor binding in *Pseudomonas aeruginosa*, along with analyses of its regulatory network, key virulence and metabolic regulators, and a pangenomic examination of transcription factors. Utilizing large-scale ChIP-seq and multi-omics integration, the research **convincingly** supports the hierarchical regulatory structures and offers insights into virulence mechanisms. This dataset, made available through an online database, should be an invaluable resource to the research community studying *P. aeruginosa*, a key pathogen at risk for hospital infections and development of antibiotic resistance.

---

## [Referee Report · Reviewer #1 (Public review)]

Summary:

In this work, Huang et al. revealed the complex regulatory functions and transcription network of 172 unknown transcriptional factors (TFs) in *Pseudomonas aeruginosa* PAO1. They have built a global TF-DNA binding landscape and elucidated binding preferences and functional roles of these TFs. More specifically, the authors established a hierarchical regulatory network and identified ternary regulatory motifs, and co-association modules. Since *P. aeruginosa* is a well known pathogen, the authors thus identified key TFs associated with virulence pathways (e.g., quorum sensing [QS], motility, biofilm formation), which could be potential drug targets for future development. The authors also explored the TF conservation and functional evolution through pan-genome and phylogenetic analyses. For the easy searching by other researchers, the authors developed a publicly accessible database (PATF_Net) integrating ChIP-seq and HT-SELEX data.

Strengths:

(1) The authors performed ChIP-seq analysis of 172 TFs (nearly half of the 373 predicted TFs in *P. aeruginosa*) and identified 81,009 significant binding peaks, representing one of the largest TF-DNA interaction studies in the field. Also, The integration of HT-SELEX, pan-genome, and phylogenetic analyses provided multi-dimensional insights into TF conservation and function.

(2) The authors provided informative analytical Framework for presenting the TFs, where a hierarchical network model based on the "hierarchy index (h)" classified TFs into top, middle, and bottom levels. They identified 13 ternary regulatory motifs and co-association clusters, which deepened our understanding of complex regulatory interactions.

(3) The PATF_Net database provides TF-target network visualization and data-sharing capabilities, offering practical utility for researchers especially for the *P. aeruginosa* field.

Weaknesses:

(1) There is very limited experimental validation for this study. Although 24 virulence-related master regulators (e.g., PA0815 regulating motility, biofilm, and QS) were identified, functional validation (e.g., gene knockout or phenotypic assays) is lacking, leaving some conclusions reliant on bioinformatic predictions. Another approach for validation is checking the mutations of these TFs from clinical strains of *P. aeruginosa*, where chronically adapted isolates often gain mutations in virulence regulators.

(2) ChIP-seq in bacteria may suffer from low-abundance TF signals and off-target effects. The functional implications of non-promoter binding peaks (e.g., coding regions) were not discussed.

(3) PATF_Net currently supports basic queries but lacks advanced tools (e.g., dynamic network modeling or cross-species comparisons). User experience and accessibility remain under-evaluated. But this could be improved in the future.

Achievement of Aims and Support for Conclusions

(1) The authors successfully mapped global *P. aeruginosa* TF binding sites, constructed hierarchical networks and co-association modules, and identified virulence-related TFs, fulfilling the primary objectives. The database and pan-genome analysis provide foundational resources for future studies.

(2) The hierarchical model aligns with known virulence mechanisms (e.g., LasR and ExsA at the bottom level directly regulating virulence genes). Co-association findings (e.g., PA2417 and PA2718 co-regulating pqsH) resonate with prior studies, though experimental confirmation of synergy is needed.

Impact on the Field and Utility of Data/Methods

(1) This study fills critical gaps in TF functional annotation in *P. aeruginosa*, offering new insights into pathogenicity mechanisms (e.g., antibiotic resistance, host adaptation). The hierarchical and co-association frameworks are transferable to other pathogens, advancing comparative studies of bacterial regulatory networks.

(2) PATF_Net enables rapid exploration of TF-target interactions, accelerating candidate regulator discovery.

Comments on revisions:

The authors have done a good job of revising their manuscript. The manuscript is now more concise and logical for readers.

---

## [Referee Report · Reviewer #3 (Public review)]

Summary:

The authors utilized ChIP-seq on strains containing tagged transcription factor (TF)-overexpression plasmids to identify binding sites for 172 transcription factors in *P. aeruginosa*. High-quality binding site data provides a rich resource for understanding regulation in this critical pathogen. These TFs were selected to fill gaps in prior studies measuring TF binding sites in *P. aeruginosa*. The authors further perform a structured analysis of the resulting transcriptional regulatory network, focusing on regulators of virulence and metabolism, in addition to performing a pangenomic analysis of the TFs. The resulting dataset has been made available through an online database. While the implemented approach to determining functional TF binding sites has limitations, the resulting dataset still has substantial value to *P. aeruginosa* research.

Strengths:

The generated TF binding site database fills an important gap in regulatory data in the key pathogen *P. aeruginosa*. Key analyses of this dataset presented include an analysis of TF interactions and regulators of virulence and metabolism, which should provide important context for future studies into these processes. Experimental validation has been included in the revised version. The online database containing this data is well organized and easy to access. As a data resource, this work should be of significant value to the infectious disease community.

Weaknesses:

Drawbacks of the study, which have been mitigated in a revised version, include (1) challenges interpreting binding site data obtained from TF overexpression due to unknown activity state of the TFs on the measured conditions (discussed by the authors), and (2) remaining challenges in the practical utilization of the TRN topological analysis.

---

## [Author Response]

The following is the authors’ response to the previous reviews

**Public Reviews:**

**Reviewer #1 (Public review):**
Summary:In this work, Huang et al. revealed the complex regulatory functions and transcription network of 172 unknown transcriptional factors (TFs) in *Pseudomonas aeruginosa* PAO1. They have built a global TF-DNA binding landscape and elucidated binding preferences and functional roles of these TFs. More specifically, the authors established a hierarchical regulatory network and identified ternary regulatory motifs, and co-association modules. Since *P. aeruginosa* is a well known pathogen, the authors thus identified key TFs associated with virulence pathways (e.g., quorum sensing [QS], motility, biofilm formation), which could be potential drug targets for future development. The authors also explored the TF conservation and functional evolution through pan-genome and phylogenetic analyses. For the easy searching by other researchers, the authors developed a publicly accessible database (PATF_Net) integrating ChIP-seq and HT-SELEX data.Strengths:(1) The authors performed ChIP-seq analysis of 172 TFs (nearly half of the 373 predicted TFs in *P. aeruginosa*) and identified 81,009 significant binding peaks, representing one of the largest TF-DNA interaction studies in the field. Also, The integration of HT-SELEX, pan-genome, and phylogenetic analyses provided multi-dimensional insights into TF conservation and function.(2) The authors provided informative analytical Framework for presenting the TFs, where a hierarchical network model based on the "hierarchy index (h)" classified TFs into top, middle, and bottom levels. They identified 13 ternary regulatory motifs and co-association clusters, which deepened our understanding of complex regulatory interactions.(3) The PATF_Net database provides TF-target network visualization and data-sharing capabilities, offering practical utility for researchers especially for the *P. aeruginosa* field.

Thank you for your positive feedback!

Weaknesses:(1) There is very limited experimental validation for this study. Although 24 virulence-related master regulators (e.g., PA0815 regulating motility, biofilm, and QS) were identified, functional validation (e.g., gene knockout or phenotypic assays) is lacking, leaving some conclusions reliant on bioinformatic predictions. Another approach for validation is checking the mutations of these TFs from clinical strains of *P. aeruginosa*, where chronically adapted isolates often gain mutations in virulence regulators.

Thank you for this valuable suggestion. We have performed the EMSA experiment to validate the binding result and also constructed the mutants for further functional validation. The details can be found in Figure S5.

(2) ChIP-seq in bacteria may suffer from low-abundance TF signals and off-target effects. The functional implications of non-promoter binding peaks (e.g., coding regions) were not discussed.

Thank you for this insightful comment regarding ChIP-seq data quality and non-promoter binding events. While we acknowledge that completely eliminating all non-specific binding signals is technically challenging in bacterial ChIP-seq experiments, we implemented stringent quality control measures including replicates, negative controls, and FDR cutoffs to minimize false positives.

Although the coding binding peaks represent a smaller fraction of total binding events, they are functionally significant rather than mere technical artifacts. Our previous work systematically demonstrated that bacterial TFs can bind to coding sequences and regulate gene expression through multiple mechanisms, including modulating cryptic promoter activity and antisense RNA transcription, hindering transcriptional elongation, and influencing translational efficiency[1]. We have now expanded the Discussion section to address these regulatory mechanisms.

(3) PATF_Net currently supports basic queries but lacks advanced tools (e.g., dynamic network modeling or cross-species comparisons). User experience and accessibility remain underevaluated. But this could be improved in the future.

Thank you for this constructive feedback on PATF_Net. We acknowledge that more advanced features would further enhance the platform’s utility. To enhance the utility of PA_TFNet, we have implemented two new features: (1) a virulence pathway browser that allows users to explore TF binding across curated gene sets for key virulence pathways (quorum sensing, secretion systems, biofilm, motility, etc.), and (2) a target gene search function that enables rapid identification of all TFs regulating any gene of interest by locus tag query.

Achievement of Aims and Support for Conclusions(1) The authors successfully mapped global *P. aeruginosa* TF binding sites, constructed hierarchical networks and co-association modules, and identified virulence-related TFs, fulfilling the primary objectives. The database and pan-genome analysis provide foundational resources for future studies.(2) The hierarchical model aligns with known virulence mechanisms (e.g., LasR and ExsA at the bottom level directly regulating virulence genes). Co-association findings (e.g., PA2417 and PA2718 co-regulating pqsH) resonate with prior studies, though experimental confirmation of synergy is needed.

Thank you for your positive feedback! We have added experimental validation in the Results section.

Impact on the Field and Utility of Data/Methods(1) This study fills critical gaps in TF functional annotation in *P. aeruginosa*, offering new insights into pathogenicity mechanisms (e.g., antibiotic resistance, host adaptation). The hierarchical and co-association frameworks are transferable to other pathogens, advancing comparative studies of bacterial regulatory networks.(2) PATF_Net enables rapid exploration of TF-target interactions, accelerating candidate regulator discovery.

Thank you for your positive feedback!

**Reviewer #3 (Public review):**
Summary:The authors utilized ChIP-seq on strains containing tagged transcription factor (TF)-overexpression plasmids to identify binding sites for 172 transcription factors in *P. aeruginosa*. High-quality binding site data provides a rich resource for understanding regulation in this critical pathogen. These TFs were selected to fill gaps in prior studies measuring TF binding sites in *P. aeruginosa*. The authors further perform a structured analysis of the resulting transcriptional regulatory network, focusing on regulators of virulence and metabolism, in addition to performing a pangenomic analysis of the TFs. The resulting dataset has been made available through an online database. While the implemented approach to determining functional TF binding sites has limitations, the resulting dataset still has substantial value to *P. aeruginosa* research.Strengths:The generated TF binding site database fills an important gap in regulatory data in the key pathogen *P. aeruginosa*. Key analyses of this dataset presented include an analysis of TF interactions and regulators of virulence and metabolism, which should provide important context for future studies into these processes. The online database containing this data is well organized and easy to access. As a data resource, this work should be of significant value to the infectious disease community.

Thank you for your positive feedback!

Weaknesses:Drawbacks of the study include (1) challenges interpreting binding site data obtained from TF overexpression due to unknown activity state of the TFs on the measured conditions, (2) limited practical value of the presented TRN topological analysis, and (3) lack of independent experimental validation of the proposed master regulators of virulence and metabolism.

We thank the reviewer for summarizing these key concerns. We acknowledge the limitations raised regarding TF overexpression, TRN topological analysis interpretation, and experimental validation. We provide detailed point-by-point responses to each of these concerns in our replies to the specific comments below, where we explain our rationale, the measures taken to address these limitations, and our plans for improvement.

**Recommendations for the authors:**

**Reviewer #1 (Recommendations for the authors):**
Future Directions for the authors to consider for next steps:(1) Key TFs (e.g., PA1380, PA5428) should be validated via gene knock out experiments, fluorescent reporter assays, or animal models to confirm roles in virulence pathways.

Thank you for this important suggestion. We agree that experimental validation is essential to confirm their regulatory roles and biological functions.

Firstly, we selected a subset of key TFs, including PA0167, PA1380, PA0815, and PA3094, and performed Electrophoretic Mobility Shift Assays (EMSA) experiments to validate their direct binding to target promoters. These results confirmed the ChIP-seq-identified interactions and are now included as Figure S5A-F.

We also constructed a clean deletion mutant of PA1380 and PA 3094 (ΔPA1380 and ΔPA3094) and their complementary strains (ΔPA1380/p and ΔPA3094/p). We then performed RT-qPCR analysis to validate their regulatory effects on key target genes. We found that PA1380 positively regulate the expression of *cupB1* and *cupB3* genes (Figure S5F). While the CupB cluster was known not be as important as CupA cluster in the biofilm information, so we did not find significant difference in biofilm formation between WT and ΔPA1380. Additionally, we found TF PA3094 also positively regulate *lecA* expression, which were shown in Figure S5G.

We agree that comprehensive functional validation, including animal model studies, would further strengthen the biological significance of these findings. Such experiments are currently underway in our laboratory and will be the subject of follow-up studies.

We have revised the Results section and Method section to include these validation experiments and their implications. Please see Figure S5 and Lines 283-300.

“To experimentally validate the regulatory interactions identified by ChIP-seq, we performed biochemical and genetic analyses on selected TFs. First, we conducted Electrophoretic Mobility Shift Assays (EMSA) for four TFs, including PA0167, PA0815, PA1380, and PA3094, using DNA fragments containing their predicted binding sites from target gene promoters. These TFs showed specific binding to their cognate DNA sequences (Figure S5A-D), confirming the direct binding of the ChIP-seq-identified interactions.

To further validate the functional regulatory roles of these TFs, we constructed clean deletion mutants of PA1380 and PA3094 (ΔPA1380 and ΔPA3094) along with their complemented strains (ΔPA1380/p and ΔPA3094/p). RT-qPCR analysis revealed that PA1380 positively regulates the expression of cupB1 and cupB3 (Figure S5E), two genes within the CupB fimbrial cluster identified as ChIP-seq targets. Similarly, PA3094 was confirmed to positively regulate lecA expression (Figure S5F), which encodes a lectin involved in biofilm formation and host interactions[2]. Expression of these target genes was restored to wild-type (WT) levels in the complemented strains, validating the regulatory relationships predicted by ChIP-seq. These combined biochemical and genetic validations demonstrate the accuracy and biological relevance of our TF binding data.”

(2) Non-promoter binding events (e.g., coding regions) may regulate RNA stability, warranting integration with translatomics or epigenomics data.

Thank you for this suggestion. We have now expanded the Discussion section to address this comment. Please see Lines 478-482.

“Our analysis revealed that TF binding events occur within coding regions, which is consistent with our previous study demonstrating that bacterial TFs possess binding capabilities for coding regions and can regulate transcription through multiple mechanisms [1]. Besides, it may also regulate RNA stability, warranting integration with translatomics or epigenomics data.”

(3) Incorporate strain-specific TF data (e.g., clinical isolates) and dynamic visualization tools to broaden PATF_Net's applicability.

Thank you for this constructive suggestion. To enhance the utility of PA_TFNet, we have implemented two new features: (1) a virulence pathway browser that allows users to explore TF binding across curated gene sets for key virulence pathways (quorum sensing, secretion systems, biofilm, motility, etc.), and (2) a target gene search function that enables rapid identification of all TFs regulating any gene of interest by locus tag query. These features are now live on the database and described in the revised manuscript.

Regarding strain-specific TF data, we agree this would be valuable for understanding regulatory diversity in clinical isolates. However, such an expansion would require ChIP-seq profiling across multiple strains. The current dataset is based on the reference strain PAO1, which serves as the foundation for most *P. aeruginosa* research and allows direct comparison with existing genomic and functional studies. We have added a statement in the revised manuscript acknowledging this limitation and highlighting strain-specific TF analysis as an important future direction for the field. Please see Lines 372-390.

“The database offers multiple search modalities to facilitate data exploration: users can perform TF-centric searches to query binding sites, target genes, and regulatory networks for individual TFs, or utilize the target gene search function to identify all TFs that regulate any gene of interest by entering its locus tag. To connect regulatory data with biological function, we have implemented a virulence pathway browser that allows users to explore TF binding patterns across curated gene sets for major *P. aeruginosa* virulence pathways. Interactive visualization tools, including network graphs and binding profile plots, facilitate intuitive exploration of regulatory relationships. The primary purpose of PATF_Net is to store, search, and mine valuable information on *P. aeruginosa* TFs for researchers investigating *P. aeruginosa* infection. The current resource is based on the reference strain PAO1, which serves as the foundation for most *P. aeruginosa* molecular studies and allows direct integration with existing genomic annotations and functional data. However, *P. aeruginosa* exhibits substantial genomic diversity across clinical isolates, and strain-specific differences in TF binding patterns may contribute to phenotypic variation in virulence, antibiotic resistance, and host adaptation. Extension of this resource to include strain-specific regulatory maps from diverse clinical isolates would provide valuable insights into the regulatory basis and represents an important direction for future investigation.”

(4) Phylogenetic analysis highlights TF conservation in bacteria; future work could explore functional homology in other Gram-negative pathogens (e.g., *E. coli*).

Thank for this insightful suggestion. Our phylogenetic analysis revealed that *P. aeruginosa* TFs exhibit varying degrees of conservation across bacterial species, with some showing broad distribution across Gram-negative pathogens while others are lineage-specific.

We agree that exploring functional homology of orthologous TFs across species would be highly valuable. Such comparative studies could address whether conserved TFs regulate similar target genes and biological processes across species, or whether regulatory networks have been rewired during evolution. For example, comparative ChIP-seq analysis of *P. aeruginosa* TFs and their orthologs in *Klebsiella pneumoniae* or even Gram-positive pathogen like *Bacillus cereus* could reveal conserved regulatory modules governing universal virulence or metabolic strategies versus species-specific adaptations. This represents an important direction for future investigation and would be facilitated by the comprehensive TF binding dataset we provide here. We have expanded the Discussion section to highlight this future direction. Please see Lines 539-550.

“While our phylogenetic analysis reveals varying degrees of TF conservation across bacterial species, the functional implications of this conservation remain to be fully explored. Many *P. aeruginosa* TFs have clear orthologs in both Gram-negative (e.g., *Klebsiella pneumoniae*) and Gram-positive pathogens (e.g., *Bacillus cereus*), yet whether these orthologs regulate similar target genes and biological processes is largely unknown. Future comparative ChIP-seq profiling of orthologous TFs could reveal the extent to which regulatory network architecture is conserved versus rewired during bacterial evolution, potentially identifying core regulatory modules governing universal bacterial strategies versus species-specific innovations. Such cross-species comparisons would enhance our understanding of regulatory network evolution and enable functional prediction in less well-characterized pathogens based on homology to experimentally validated *P. aeruginosa* regulators.”

**Reviewer #3 (Recommendations for the authors):**
Major comments- Limitations of the ChIP-seq approach: With overexpression plasmids as an approach to TRN elucidation, there are always a set of concerns. First, TF expression is not enough to ensure regulatory activity - metabolite effects must be such that the TF is active which requires growing the cells in activating conditions. Second, the presence of a binding event does not mean that the binding has a regulatory effect - the authors are clearly aware of this as they specify binding sites in promoter regions, which should be helpful, but they also mention the possibility of regulatory binding events in coding regions. These issues should be listed as weaknesses of the approach in the Discussion.

Thank you for these important suggestions. We agree that these limitations should be explicitly discussed. We have now added a dedicated paragraph in the Discussion section addressing these concerns. Please see Lines 492-501.

“However, several limitations of the ChIP-seq approach should be acknowledged. Firstly, TF overexpression ensures sufficient protein levels for ChIP-seq signal detection but does not guarantee that all TFs are in their active conformational states, as many bacterial TFs require allosteric activation by metabolites, cofactors, or post-translational modifications. The cells under standard laboratory conditions which may not activate all TFs to their maximal regulatory states, potentially leading to underestimation of condition-specific binding peaks. Secondly, while we observed TF binding at thousands of genomic sites, binding per se does not equate to functional regulation, as chromatin context, cofactor availability, and competitive binding all influence regulatory outcomes.”

- Lack of independent validation: The study seems to lack substantial independent validation of either the functional nature of the binding sites as well as the proposed physiological regulatory role of the TFs. For example, for the 103 identified TF motifs, do any of these agree with existing motifs in motif databases that may be homologous to *P. aeruginosa* TFs? The authors claim to have discovered master regulators of virulence and associated core regulatory clusters - but there does not seem to be any independent validation of the proposed associations. The authors selected the TF targets to cover TFs that had not yet been characterized; however, it would have been nice to have some overlap with previous studies so that consistency and data quality could be assessed.

Thank you for raising these critical points about validation.

As for motif validation, we compared the existing motifs in the RegPrecise database[3] and we found that the motif of PA3587 show significant similarity to homologous TFs in Pseudomonadaceae. We have added the related description in the Results section. Please see Figure S3B and Lines 228-231.

As for the validation of master regulators, we have performed EMSA experiments for validating the binding events and constructed the mutants for function validation. We have added the related contents in Results section. Please see Figure S5 and Lines 283-300.

We have discussed the overlap between our results and previous studies in the Discussion section. Please see Lines 530-538.

“PA0797 is known to regulate the pqs system and pyocyanin production[4]. In the present study, it was also found to bind to the pqsH promoter region and its motif was visualised. PA5428 was found to bind to the promoter regions of aceA and glcB genes[5], which was also demonstrated in our ChIP-seq results. PA4381 (CloR) was found to be associated with polymyxin resistance in a previous study[6] and to be possibly related to ROS resistance in the present study. Furthermore, PA5032 plays a putative role in biofilm regulation and also forms an operon with PA5033, an HP associated with biofilm formation[7].”

- Uncertain value of TRN topology analysis: The relationship between ternary motifs and pathogenicity of *P. aeruginosa*, and why the authors argue these results motivated TF-targeting drugs (the topic of the last paragraph of the Discussion), are unclear to me. The authors allude to possible connections between pathogenicity, growth, and drug resistance, but I don't see concrete examples here of related TF interactions that clearly represent these relationships. The sections "Hierarchical networks of TFs based on pairwise interactions" and "Ternary regulatory motifs show flexible relationships among TFs in *P. aeruginosa*" seem to not say much in terms of results that are actionable or possible to validate. A topological graph is constructed based on observed TF-TF connections in measured binding sites - however, it's unclear if any of these connections are physiologically meaningful. Line 178 - Why would there be any connection between the structural family of TF and its location in the proposed TRN hierarchy?

Thank you for this valuable comment on TRN topology analysis. It is hard to quantify precisely how much this resource will accelerate *P. aeruginosa* research or drug development, but we believe providing this foundational network architecture has inherent value for the community, which is valued for enabling hypothesis generation even before comprehensive functional validation. We would like to clarify our perspective on these findings and have added the discussion in the revised manuscript to better describe their nature and value. Please see Lines 517-528.

“Additionally, although the TRN analysis revealed organizational patterns in *P. aeruginosa* regulatory network, the functional significance these topological features, including their specific contributions to pathogenicity, metabolic adaptation, and antibiotic resistance remains to be experimentally determined in the future work. The hierarchical structure and regulatory motifs we identified represent objective network properties derived from our binding data, but translating these structural observations into mechanistic understanding will require condition-specific functional studies, genetic validation, and phenotypic characterization. Our analysis provided a systematic framework and generating testable hypotheses rather than definitive functional conclusions. Nevertheless, these network-level organizational principles provided value to the community as a foundational reference, similar to other regulatory network maps[8] that were useful even before comprehensive validation.”

- Identification of "master" regulators: Line 527 on virulence regulators: "We first generated gene lists associated with nine pathways" - is this not somewhat circular, i.e. using gene lists generated from (I assume) co-regulated gene sets to identify regulators of those gene lists? I can't tell from the cited reference (80), which is their own prior review article, what the original source of these gene lists was. Somewhat related to this point - Line 32: 24 "master regulators" - if there are that many, is it still considered a master regulator? Line 270: This term "master regulator" would seem to require some quantitative justification. Identifying 24 (a large number of) "master" regulators of virulence would seem to dilute the implied power of the term.

We apologize for the lack of clarity regarding the virulence pathway gene lists, and we have provided complete gene lists for virulence-related pathways, which were compiled from functional annotations, in our online PA_TFNet database.

Additionally, we appreciate your concern about the use of “master” regulator. The usage is based on previous studies[9,10], and the master regulator is commonly known in the development of multicellular organisms as a subset of TFs that control the expression of multiple downstream genes and govern lineage commitment or key biological processes. We employed the term "master regulator" in an analogous manner to specify a class of functionally crucial TFs that participate in a pathway or biological event by regulating multiple downstream genes statistically enriched in that pathway. In line with this definition, we identified TFs whose targets were significantly enriched in genes associated with specific virulence pathways (hypergeometric test, P < 0.05).

We understand the concern that identifying 24 master regulators might seem to dilute the term. However, we would like to clarify that each of these 24 TFs is a "master regulator" with respect to specific virulence pathways based on statistical criteria, not necessarily a global master regulator of multiple pathways of *P. aeruginosa*. We have revised the Method section. Please see Lines 604-612.

- Line 234: "Genome-wide synergistic co-association of TFs in *P. aeruginosa*." This section was an interesting analysis. As I mention above, the weakness of an overexpression approach is not knowing whether the TF is active on the examined conditions. By looking at shared binding peaks across overexpression of different TFs, it should indeed be possible to glean some regulatory connections across TFs. Furthermore, the authors discuss specific examples that appear physiologically reasonable, which is appreciated.

We thank the reviewer for this positive assessment of our co-association analysis. We agree with the limitation of the overexpression approach, which have been discussed in the Discussion section. We are pleased that the reviewer found the approach and specific examples valuable.

Minor comments- Line 35 - "high-throughput systematic evolution of ligands by exponential enrichment" - no idea what this means. Is this related to the web-based database, or why is it mentioned in the same sentence?

We apologize for the unclear presentation. To clarify: “High-throughput systematic evolution of ligands by exponential enrichment” (HT-SELEX) is an in vitro technique for determining TF DNA-binding motifs, which our group previously applied to a subset of *P. aeruginosa* TFs in a prior publication[11]. In the current study, we performed ChIP-seq for 172 TFs, which represent the majority of TFs not covered by the previous HT-SELEX study. Together, these two complementary approaches (HT-SELEX for in vitro binding motifs, ChIP-seq for in vivo genomic binding sites) provide near-complete coverage of the *P. aeruginosa* TF repertoire. Both datasets are integrated into our PA_TFNet database.

Due to space constraints in the abstract, we could not provide detailed explanation of HT-SELEX, but we have now improved the clarity in the Introduction to better explain the relationship between our previous HT-SELEX work and the current ChIP-seq study, and why both are mentioned together in the context of the database. Please see Lines 99-105.

- Line 193 - Only 9 auto-regulating TFs seems like a low number, given the frequency of negative auto-regulation in other organisms like *E. coli*. Could the authors comment on their expectations based on well-curated TRNs?

Thank you for this comment. We agree that 9 auto-regulating TFs is lower than might be expected based on *E. coli*, where auto-regulation is more prevalent. This likely reflects technical limitations of ChIP-seq approach that our detection was limited to standard growth conditions rather than the diverse physiological states where auto-regulation often occurs. Therefore, the 9 TFs we report represent a high-confidence subset, and the true frequency of auto-regulation in *P. aeruginosa* likely is higher. We added the content in the revised manuscript. Please see Lines 193-196.

“This number likely represents a conservative estimate, as experiments may not optimally capture auto-regulatory events that depend on native expression levels or specific physiological conditions.”

- Line 230 - "This conservation suggests that TFs within the same cluster co-regulate similar sets of genes." - Why would clustering of TF binding site motifs need to be done to make this assessment? Couldn't the shared set of regulated genes be identified directly from the binding site data? Computing TF binding site motifs has obvious value, but I am struggling to understand the point of clustering the motifs. Is there some implied evolutionary or physiological connection here? No specific physiological roles or hypotheses are discussed in this section.

Thank you for this important question. We agree that shared target genes can be identified directly from ChIP-seq binding data, which we also analyzed (co-association analysis). The motif clustering analysis serves a complementary and distinct purpose that provides information not directly obtainable from overlapped targets alone. Specifically, target overlap is inherently condition dependent, and motif clustering captures this intrinsic binding specificity, which reflects the structural similarity of DBDs, evolutionary relationships, and potential for functional redundancy or cooperativity under specific conditions. We have revised the related content in the manuscript, and please see Lines 236-242.

“Clustering of TF binding motifs identified groups of TFs with similar intrinsic DNA-binding specificities. As expected, many clusters contained TFs from the same DBD families, reflecting evolutionary conservation and potential functional redundancy or competitive binding at shared regulatory elements. Notably, the clustering also uncovered associations between TFs from different DBD families, suggesting convergent evolution of binding specificity or novel regulatory interactions that warrant further investigation.”

- Line 284 - should "metabolomic" be "metabolic"? I didn't see metabolomic data

Yes, we have revised. Please see Line 311.

- Several of the figures are too small (e.g. Fig S4A) or complex (Fig 2A) to see clearly or glean information from.

Thank you for this comment. We acknowledge that Figure 2A and Figure S4A contain dense information due to the comprehensive nature of the regulatory network and the large number of TFs analyzed. We believe these overview figures serve an important purpose in conveying the scale and organization of the regulatory network, while the tables (Table S6 for Fig. S4A and Table S3 for Fig. 2A) provide the granular data needed for specific inquiries. We have also made the figures available in higher resolution and increased font sizes where possible without compromising the overall layout.

- I don't understand the organization of the "Ternary regulatory motifs" in Supplementary Data File 4 - A table of contents explaining the tabs and columns would be welcome (for this as well as other supplementary files, some of which are more straightforward than others).

Thank you for this suggestion. We have now revised all supplementary data files to include header and necessary annotations in the first row. Specifically for Supplementary Data File 4, the three columns (Top, Middle, Bottom) represent the left, middle, and right node, respectively, in each ternary regulatory motif.

- I would have expected genomic locations of TF binding sites would have been one of the Supplementary Tables, to increase the accessibility of the data. However, the data is made available through their website, https://jiadhuang0417.shinyapps.io/PATF_Net/, which was easy to access and download the full dataset, so this is a minor issue.

Thank for accessing our PA_TFNet database and for the positive feedback on data accessibility. We agree that providing genomic locations of TF binding sites is crucial. These data are fully available and downloadable through the web interface, which allows flexible searching, filtering, and batch download of binding sites. We felt that the interactive and database format provides more functionality than static supplementary tables (e.g., dynamic filtering by TF, genomic region, or binding strength), given the large scale of this dataset.

References

(1) Hua, C., Huang, J., Wang, T., Sun, Y., Liu, J., Huang, L. et al. Bacterial Transcription Factors Bind to Coding Regions and Regulate Internal Cryptic Promoters. Mbio 13, e0164322 (2022).

(2) Chemani, C., Imberty, A., de Bentzmann, S., Pierre, M., Wimmerová, M., Guery, B. P. et al. Role of LecA and LecB lectins in *Pseudomonas aeruginosa*-induced lung injury and effect of carbohydrate ligands. Infect Immun 77, 2065-2075 (2009).

(3) Novichkov, P. S., Kazakov, A. E., Ravcheev, D. A., Leyn, S. A., Kovaleva, G. Y., Sutormin, R. A. et al. RegPrecise 3.0–a resource for genome-scale exploration of transcriptional regulation in bacteria. Bmc Genomics 14, 745 (2013).

(4) Cui, G. Y., Zhang, Y. X., Xu, X. J., Liu, Y. Y., Li, Z., Wu, M. et al. PmiR senses 2-methylisocitrate levels to regulate bacterial virulence in *Pseudomonas aeruginosa*. Sci Adv 8 (2022).

(5) Hwang, W., Yong, J. H., Min, K. B., Lee, K.-M., Pascoe, B., Sheppard, S. K. et al. Genome-wide association study of signature genetic alterations among *Pseudomonas aeruginosa* cystic fibrosis isolates. Plos Pathog 17, e1009681 (2021).

(6) Gutu, A. D., Sgambati, N., Strasbourger, P., Brannon, M. K., Jacobs, M. A., Haugen, E. et al. Polymyxin resistance of *Pseudomonas aeruginosa* phoQ mutants is dependent on additional two-component regulatory systems. Antimicrob Agents Chemother 57, 2204-2215 (2013).

(7) Zhang, L., Fritsch, M., Hammond, L., Landreville, R., Slatculescu, C., Colavita, A. et al. Identification of genes involved in *Pseudomonas aeruginosa* biofilm-specific resistance to antibiotics. PLoS One 8, e61625 (2013).

(8) Galan-Vasquez, E., Luna, B. & Martinez-Antonio, A. The Regulatory Network of *Pseudomonas aeruginosa*. Microb Inform Exp 1, 3 (2011).

(9) Fan, L. G., Wang, T. T., Hua, C. F., Sun, W. J., Li, X. Y., Grunwald, L. et al. A compendium of DNA-binding specificities of transcription factors in Pseudomonas syringae. Nat Commun 11 (2020).

(10) Chan, S. S.-K. & Kyba, M. What is a master regulator? Journal of stem cell research & therapy 3, 114 (2013).

(11) Wang, T. T., Sun, W. J., Fan, L. G., Hua, C. F., Wu, N., Fan, S. R. et al. An atlas of the binding specificities of transcription factors in *Pseudomonas aeruginosa* directs prediction of novel regulators in virulence. Elife 10 (2021).